# Distinct effects of prefrontal and parietal cortex inactivations on an accumulation of evidence task in the rat

Jeffrey C Erlich[1,2]*, Bingni W Brunton[2,3], Chunyu A Duan[2], Timothy D Hanks[2], Carlos D Brody[2,4]*

[1]NYU-ECNU Institute of Brain and Cognitive Science, NYU Shanghai, Shanghai, China; [2]Princeton Neuroscience Institute, Department of Molecular Biology, Princeton University, Princeton, United States; [3]Department of Biology, UW Institute of Neuroengineering, University of Washington, Seattle, United States; [4]Howard Hughes Medical Institute, Princeton University, Princeton, United States

**Abstract** Numerous brain regions have been shown to have neural correlates of gradually accumulating evidence for decision-making, but the causal roles of these regions in decisions driven by accumulation of evidence have yet to be determined. Here, in rats performing an auditory evidence accumulation task, we inactivated the frontal orienting fields (FOF) and posterior parietal cortex (PPC), two rat cortical regions that have neural correlates of accumulating evidence and that have been proposed as central to decision-making. We used a detailed model of the decision process to analyze the effect of inactivations. Inactivation of the FOF induced substantial performance impairments that were quantitatively best described as an impairment in the output pathway of an evidence accumulator with a long integration time constant (>240 ms). In contrast, we found a minimal role for PPC in decisions guided by accumulating auditory evidence, even while finding a strong role for PPC in internally-guided decisions.

*For correspondence: jerlich@ nyu.edu (JCE); brody@princeton. edu (CDB)

**Competing interests:** The authors declare that no competing interests exist.

## Introduction

Gradual accumulation of evidence for or against different choices has been implicated in many types of decision-making, including value-based decisions (*Basten et al., 2010*; *Milosavljevic et al., 2010*; *Cavanagh et al., 2011*; *Hunt et al., 2012*; *Solway and Botvinick, 2012*), social decisions (*Krajbich and Rangel, 2011*), economic decisions (*Gluth et al., 2012*), gambling decisions (*Busemeyer and Townsend, 1993*), memory-based decisions (*Ratcliff, 1978*), numerical comparison decisions (*Sigman and Dehaene, 2005*), visual search decisions (*Purcell et al., 2010*; *Heitz and Schall, 2012*), and perceptual (*Gold and Shadlen, 2007*; *Ratcliff et al., 2007*; *Mante et al., 2013*) decisions. It is therefore considered a core decision-making process. Although neural correlates of evidence accumulation have been reported in several interconnected primate brain regions—such as PPC (*Shadlen and Newsome, 2001*; *Roitman and Shadlen, 2002*; *Hunt et al., 2012*), prefrontal cortex (*Hunt et al., 2012*) including frontal eye fields (FEF; *Kim and Shadlen, 1999*; *Purcell et al., 2010*; *Ding and Gold, 2012*; *Heitz and Schall, 2012*; *Mante et al., 2013*), striatum (*Ding and Gold, 2010*), and superior colliculus (*Horwitz and Newsome, 1999*; *Ratcliff et al., 2007*)—the specific roles of these different brain regions in decisions driven by accumulation of evidence have not yet been distinguished.

We recently developed a rat model of gradual accumulation of evidence for decision-making, using a task that allows detailed quantitative modeling of the accumulation and decision processes ('Poisson Clicks' task; *Brunton et al., 2013*). In separate work from our laboratory using the Poisson Clicks task, electrophysiological recordings in rat PPC and Frontal Orienting

**eLife digest** Imagine that you have to buy a computer before the start of the school year. You have a few options, such as a laptop or a desktop, each with its own advantages and disadvantages. A laptop is relatively light and portable, whereas a desktop has more memory and is cheaper. You will gradually accumulate evidence for and against each option, but before school starts, you have to make a decision.

This gradual accumulation of evidence is an important element in many forms of decision making. It is known that the activity of many regions within the brain seem to represent accumulation of evidence, but relatively little is known about the causal role played by each region in the decision-making process. Now, by performing a series of experiments on rats, Erlich et al. have clarified the precise roles of two of these regions: the frontal orienting fields in prefrontal cortex and the posterior parietal cortex. In the experiments the rats listened to a series of clicks from two speakers, one to the left and one to the right, and then had to decide if more clicks came from the left or the right speaker.

The rats normally used all of the accumulated evidence (up to 1 second) for their decision. When the posterior parietal cortex was silenced (using a drug called muscimol), the rats continued to use all of the evidence available to them. However, when the frontal orienting fields were silenced (again using muscimol), decisions were driven only by evidence accumulated over the most recent past (just a few hundred milliseconds). So in the computer example, without the help of the frontal orienting fields, you would choose the laptop if the most recent piece of evidence was for the laptop, even if older evidence argued strongly for the desktop.

These results show that the frontal orienting fields are necessary for making decisions based on accumulated evidence, but further experiments suggested that the accumulation process itself seems to happen elsewhere in the brain. Another set of related experiments showed that the posterior parietal cortex is involved in a different type of decision making, namely 'free choice' decisions in which the rat decides between two options when there is no correct answer, such as picking a cookie from a pile of identical cookies.

Fields (FOF; *Erlich et al., 2011*) revealed classic neural correlates of evidence accumulation (*Figure 1* of *Hanks et al., 2015*). Specifically, we found neurons in these rat regions that ramp up their activity during the stimulus, and the slope of that ramp is correlated with the strength of the momentary evidence just as one would expect from neurons whose firing rates represent the accumulation of evidence over time, and just as previously reported in monkey regions that have been suggested as analogous to the rat PPC and FOF (primate PPC: *Shadlen and Newsome, 1996*, *2001*; *Roitman and Shadlen, 2002*; and monkey FEF: *Ding and Gold, 2012*; *Mante et al., 2013*; for PPC analogy, see *Whitlock et al., 2008*; *Reep and Corwin, 2009*; *Wilber et al., 2014*; for FOF/FEF analogy see *Erlich et al., 2011*).

In addition to having neural correlates of accumulating evidence (*Hanks et al., 2015*), several properties of the rat FOF suggest it as a candidate for a causal role in decisions driven by accumulation of evidence. Accumulation of evidence involves both maintaining a memory of evidence accrued so far and addition of new evidence to the memory, and is therefore linked to short-term memory processes. The rat FOF has delay activity that correlates with short-term memory, and plays a causal role in short-term memory for future orienting responses (*Erlich et al., 2011*). Furthermore, the rat FOF is well-situated to play an important role in perceptual decision-making, since it receives inputs from multiple sensory cortices (*Condé et al., 1995*), and it projects to the superior colliculus (SC; *Stuesse and Newman, 1990*), a subcortical region that, in both rodents and primates, is involved in controlling orienting motions (*Isa and Sasaki, 2002*; *Felsen and Mainen, 2008*) and is thought to be involved in decisions reported through such orienting motions. Moreover, the rat FOF is reciprocally connected with the rat PPC (*Reep et al., 1987*, *1994*), which is currently considered a critical, central node in rodent perceptual decision-making (*Carandini and Churchland, 2013*). The rodent PPC itself also has neural correlates of accumulating evidence (*Hanks et al., 2015*), and it shares with the FOF some of the key properties that suggest a causal role in decisions driven by accumulation of evidence. The rodent PPC has delay activity that correlates with performance on short-term memory tasks (*Nakamura, 1999*; *Harvey et al., 2012*), and, as shown through inactivations, plays a causal

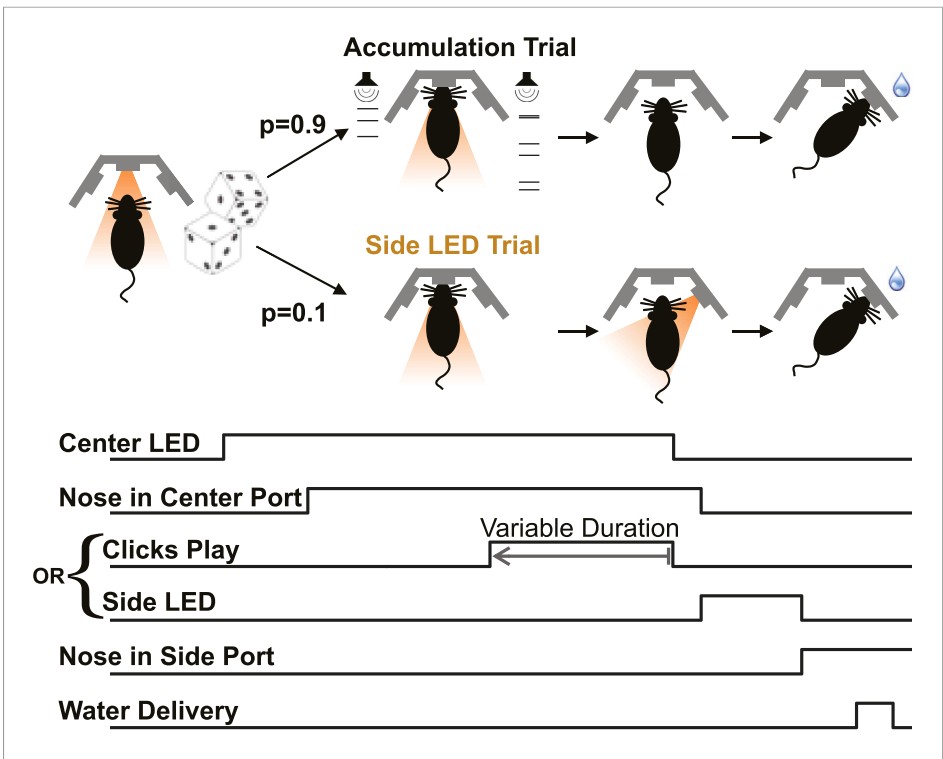

**Figure 1**. Poisson clicks accumulation task trials and interleaved side LED trials. Each accumulation task trial begins with the onset of the center LED, which signals to the rat to enter the center port. The subject holds his nose in the center port for 2 s, until the center LED offset, which is the go cue. The majority of trials (90%) are accumulation trials. On accumulation trials, clicks play from the right and left speakers (right + left click rate = 40 clicks/s), terminating with the go cue. After the go cue reward is available at the side port associated with the greater number of clicks. The stimulus duration on each trial is set by the experimenter to be in the range 0.1–1 s. On Side LED trials, no sound is played during the fixation period and one of the side ports is illuminated once the rat withdraws from the center port to indicate that reward is available there. Accumulation and side LED trials are randomly interleaved, as are left and right trials.

role in short-term memory for orienting acts (*Harvey et al., 2012*) and like FOF receives input from many sensory cortices as well as top-down input from prefrontal cortex, including FOF (*Wilber et al., 2014*). For these reasons, the FOF and the PPC are the most prominent candidate regions in rodent association cortex for being important nodes in orienting decisions guided by accumulation of evidence. We focus on these two areas here.

We implanted bilateral cannula in both FOF and PPC of rats trained to perform the Poisson Clicks task, and inactivated these regions with the GABA-A agonist muscimol while the rats performed the task. Consistent with expectations drawn from neural correlates in the rat FOF, inactivation of the FOF impaired performance in the task. We used quantitative modeling to characterize which aspect of the accumulation and decision process was impacted by inactivation of the FOF. The results of these analyses revealed a specific location for the FOF in the causal circuit underlying the Poisson Clicks behavior: the behavioral impairment caused by FOF infusions could be parsimoniously and quantitatively explained as an impairment in the premotor output pathway of an evidence accumulator with a long accumulation time constant (240 ms or more). It is possible that the decision itself (i.e., the categorization of the graded accumulator value into a discrete choice, which is a process subsequent to graded evidence accumulation) could occur in the FOF.

In contrast, we found that PPC inactivations had a relatively minor effect on the Poisson Clicks task. This was true even while the same PPC inactivations had strong effects on interleaved 'free-choice' trials, in which no sensory evidence was provided and rats were rewarded regardless of their choice of response. Our data thus suggest that the PPC plays a minimal causal role in decisions guided by accumulation of auditory evidence, while playing an important role in internally-guided decisions.

Together, our findings from inactivations of the PPC and the FOF provide important constraints on the neural circuitry underlying decisions guided by accumulation of auditory evidence in the rat.

## Results

### Behavior

We trained male Long-Evans rats (n = 14 rats) on the Poisson Clicks accumulation task (*Figure 1*, *Brunton et al., 2013*). On each trial of this task, illumination of the center LED indicated that the rat should place its nose in the center port and remain there while click trains with Poisson-generated inter-click-intervals were played from the left and right speakers. The rats learned to report which side had played the greater total number of clicks by nose-poking into the corresponding side port (*Figure 2A*). We refer to these trials as 'accumulation trials'.

In order to control for motor effects of inactivations, the accumulation trials were randomly interleaved, in most sessions, with trials that we refer to as 'side LED' trials. On side LED trials no sounds were played during fixation. Immediately after the end of fixation, one of the two side ports was illuminated, indicating availability of reward at the lit port (*Figure 1*). The right and left side LED trials, together, comprised ≈10% of the total trials.

To demonstrate that subjects accumulated the sensory evidence provided by the auditory clicks, we fit an accumulator model using the individual click times and the rats' choices on each trial (*Figure 2—figure supplement 1*; see also *Brunton et al., 2013*). Different parameter value regimes of this model can implement many different strategies, such as responding based on the first few clicks, or last few clicks, or to a burst of clicks, and many others. Consistent with previous results, maximum likelihood fits resulted in best-fit parameters associated with a gradual evidence accumulation strategy. Most importantly for this study, this strategy was characterized by a long accumulator time-constant, just under 1 s (*Figure 2B*, *Table 1*), which is the duration of the longest stimuli used here. As expected for a gradual accumulation strategy in which clicks from the entire stimulus are weighted equally, performance improved for longer stimuli with the same underlying click rates (*Figure 2C*; *Ratcliff and Rouder, 1998*; *Usher and McClelland, 2001*; *Brunton et al., 2013*), and a psychophysical reverse correlation analysis (*Kiani et al., 2008*; *Raposo et al., 2012*; *Brunton et al., 2013*) indicates that rats used clicks from all times of the stimuli to make their decision (*Figure 2D*).

### Inactivations

We report the results of five different types of inactivations: unilateral FOF, bilateral FOF, unilateral PPC, bilateral PPC, and combined bilateral FOF + unilateral PPC inactivations, for a total of 26,521 trials from 161 infusions into the FOF and PPC of 14 rats (*Figure 3A* and *Figure 3—figure supplement 1*). We initially performed muscimol inactivations of the FOF and PPC in 6 rats performing the Poisson Clicks Task (group 1). In order to verify the results and perform follow-up and control experiments, we performed inactivations in two further groups (group 2, n = 4; group 3, n = 4). The specific order and outcome of the infusions in each rat is shown in *Figure 3—figure supplement 2*.

### FOF inactivations

We placed cannulae in the center of the location currently identified as FOF (+2.0 mm AP, ±1.3 mm ML from Bregma, *Figure 3—figure supplement 1A*). These are the same coordinates used by *Erlich et al. (2011)*, and are also the coordinates at which neural correlates of accumulation of evidence were observed in the Poisson Clicks task (*Hanks et al., 2015*). For the first bilateral FOF inactivation session we infused 300 ng of muscimol per side, for a total of 600 ng. After these infusions rats did not perform the task. We subsequently used a smaller dose of 75 ng per side (Note: this is half the dose used in the bilateral PPC experiment described below). This resulted in a significant 10.3% decrement on performance on accumulation trials (p = 0.018, GLMM test; *Figure 3B*). The effect was individually significant in 3/4 rats (*Figure 3—figure supplement 3A*). Side LED trials were unimpaired (p > 0.5; *Figure 3B*), indicating that the impairment on accumulation trials was not simply a motor effect.

Unilateral infusions of muscimol into the FOF resulted in a profound bias towards ipsilateral responses in the Poisson Clicks task (*Figure 3C*). Averaged across all unilateral FOF infusion sessions, the ipsilateral bias (defined as ipsilateral % correct − contralateral % correct), was 52 ± 7% (mean ± s.e.) for accumulation trials (t-test across 12 rats $t_{11}$ = 7.27, p < $10^{-4}$; two rats in group 3 failed to perform sufficient numbers of trials during FOF inactivations to be included in this analysis). Unilateral FOF

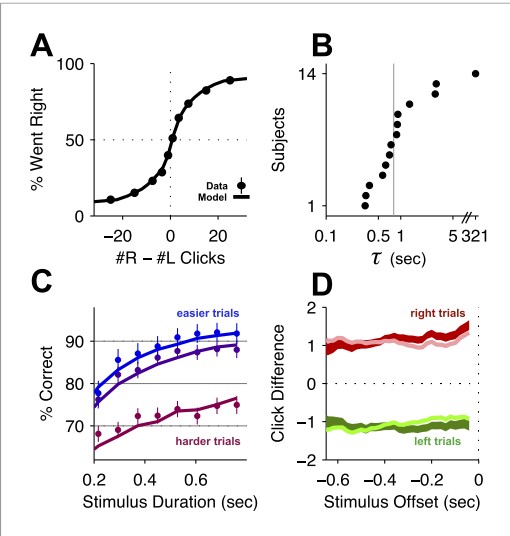

**Figure 2**. Behavioral evidence of accumulation. (**A**) Behavior as a function of total right minus total left clicks. For very easy trials (large click differences) performance is ≈90% correct. The circles (with very small error bars) are the mean ±95% binomial confidence intervals across accumulator trials from all rats 1 day before an infusion session (n = 47,580 trials across 14 rats). The thick line is the psychometric curve generated by the accumulator model fit to these trials. (**B**) The time-constant of accumulation as fit by the model for each rat in the experiment. The median (810 ms) is marked by a thin gray line. (**C**) Chronometric plot generated using the same data as in panel (**A**). The rats' performance increases with longer duration stimuli, consistent with an accumulation strategy. The circles and error bars are the mean ±95% binomial confidence intervals across trials on the easiest (blue), middle (purple) and hardest (magenta) thirds of trials defined by the absolute value of the ratios of left vs right click rates. The thick lines are the model generated chronometric curves. (**D**) Reverse correlation analyses showing that clicks throughout the stimulus were used in the rats' decision process, supporting the long accumulation time constants in (**B**). The thick dark red and green lines are the means ± std. err. across trials for where the rats went right and left. Thin light red and green lines are the model generated reverse correlation.

The following figure supplements are available for figure 2:

**Figure supplement 1**. 9-parameter Accumulator Model (reproduced from *Brunton et al., 2013*).

**Figure supplement 2**. Behavioral evidence of accumulation in individual rats.

infusions reduced performance to chance on even the easiest contralateral accumulation trials (*Figure 3C*; green data points for #R − #L ≫ 0, and red data points for #R − #L ≪ 0). The ipsilateral bias induced by FOF inactivation was highly reproducible: 87% (26/30) of individual infusion sessions resulted in a positive ipsilateral bias (sign-test, p < 0.001), and in every single rat there were more rightward responses after rightward infusions than leftward infusions (*Figure 3—figure supplement 3B*).

Importantly, as in the bilateral inactivations, there was no significant effect on side LED trials (t-test $t_5$ = 1.55, p > 0.15; *Figure 3C*, side LED Trials), nor did unilateral FOF inactivations have an effect on the response time on side LED trials (repeated-measures ANOVA, F(1,3) = 0.65, p > 0.4). This indicates that the effect of FOF inactivation was not simply an overall motor effect. Furthermore, the inactivations produced no observable effects outside of the behavioral task. Infused animals appeared normal in their home cages both immediately after the infusion and after the behavioral session. Our localized inactivations thus contrast with previous literature, in which large permanent unilateral lesions of the rat prefrontal cortex (including but extending well beyond the FOF), produced persistent, ipsiversive circling in the lesioned animals (*Crowne and Pathria, 1982*).

## Bilateral FOF inactivations reduce the subjects' accumulation time constant

Which aspects of the evidence accumulation and decision process were impaired by the FOF inactivations? To address this question, we took advantage of the accumulator model of *Brunton et al. (2013)*, which uses the knowledge the precise time of each click in each individual trial, as well as the rat's decision on each trial, to estimate 9 parameters that characterize the accumulation and decision processes (*Figure 2—figure supplement 1*). Each parameter quantifies a specific aspect of the decision process. For example, $\tau$, the time constant of the accumulator (also described by the parameter $\lambda = 1/\tau$), characterizes the time period over which the subject accumulates evidence. A negative value of $\lambda$ indicates a leaky accumulator, a positive value of $\lambda$ indicates an unstable accumulator, and perfect accumulation would have $\tau = \infty$

(i.e., $\lambda = 0$). Another example parameter is the lapse rate, which quantifies the fraction of trials in which the subject behaves as if it had ignored the clicks that were played and had instead made its decision randomly. Deviations from the perfect values ($\lambda = 0$, lapse = 0) in either of these parameters can give rise to psychometric curves with shallow slopes, and both types of imperfections can

**Table 1.** Best-fit parameters

| Ratname | $\lambda$ | $\sigma_a^2$ | $\sigma_s^2$ | $\sigma_{init}^2$ | $B$ | $\phi$ | $\tau_\phi$ | Þ | *lapse* |
|---|---|---|---|---|---|---|---|---|---|
| B115 | 1.409 | 0.113 | 102.130 | 0.523 | 14.849 | 0.175 | 0.064 | 0.157 | 0.094 |
| T055 | 1.226 | 0.001 | 11.248 | 0.043 | 16.014 | 0.253 | 0.351 | 0.118 | 0.078 |
| T057 | 0.810 | 0.031 | 74.478 | 0.027 | 15.060 | 0.156 | 0.093 | 0.020 | 0.075 |
| T058 | 1.087 | 0.000 | 17.612 | 0.000 | 15.875 | 0.025 | 0.276 | −0.122 | 0.051 |
| T061 | 0.620 | 0.000 | 96.545 | 0.502 | 16.038 | 0.380 | 0.041 | 0.236 | 0.066 |
| T062 | −0.098 | 0.000 | 49.361 | 0.619 | 15.761 | 0.139 | 0.047 | 0.518 | 0.083 |
| A065 | 2.047 | 0.000 | 37.685 | 0.207 | 15.729 | 0.147 | 0.092 | −0.465 | 0.031 |
| A066 | 0.349 | 0.000 | 15.565 | 0.000 | 12.705 | 0.072 | 0.462 | 0.041 | 0.170 |
| A077 | −2.739 | 0.197 | 128.586 | 22.801 | 9.253 | 0.184 | 0.031 | 0.886 | 0.001 |
| A078 | −2.070 | 0.000 | 104.688 | 0.000 | 18.086 | 0.283 | 0.026 | 0.062 | 0.063 |
| A060 | −1.542 | 0.000 | 54.786 | 0.000 | 15.416 | 0.010 | 0.115 | 0.180 | 0.245 |
| A062 | 2.258 | 0.296 | 156.860 | 0.486 | 16.839 | 0.527 | 0.076 | 0.466 | 0.119 |
| A083 | −0.790 | 47.441 | 31.788 | 1.384 | 16.282 | 0.015 | 0.059 | 0.033 | 0.107 |
| A084 | 1.371 | 0.064 | 70.267 | 1.690 | 15.011 | 0.016 | 0.086 | 0.467 | 0.110 |
| Meta-Rat | 1.227 | 0.001 | 57.614 | 0.043 | 16.042 | 0.221 | 0.109 | 0.065 | 0.102 |
| BiFOF | −4.144* | 62.423 | 237.642 | 1.754 | 22.013 | 0.082 | 0.039 | 0.737* | 0.010 |
| BiPPC | 1.331 | 0.531 | 42.175 | 0.000 | 14.860 | 0.512 | 0.175 | −0.249 | 0.321 |

This table shows the values of the parameters which maximize the likelihood of the full 9-parameter accumulator model for each rat, as well as for the 'meta-rat' (made from taking all of the control days that were 1 day before an infusion, n = 47,580 trials), the fit to the bilateral FOF data (n = 1809), and the fit to the bilateral PPC data (n = 1569). *indicate parameters that were significantly different from the control 'Meta-Rat'.

produce curves qualitatively similar to the experimental curve obtained after bilateral FOF inactivation (*Figure 4A*). But the similarity between the psychometric curves for the two imperfections is partly due to the fact that these psychometric curves ignore the specific timing of individual clicks. In contrast, because the two imperfections would have very different signatures in terms of how clicks at different times affect the rats' decisions, and click timing is fully taken into account in the behavioral model, the model can clearly distinguish the two imperfections. Using the control data, the best-fitting parameter values for the behavioral model had $\tau \approx 0.8$ s–indicating that subjects accumulated information over almost the entire stimulus duration but were on average slightly unstable–and a lapse rate of $\approx 0.1$ (black cross, *Figure 4B*). For data from bilateral inactivation sessions, the maximum likelihood parameter values (center of blue likelihood peak, *Figure 4B*) changed significantly. The inactivation data now had a dramatically different and much shorter accumulation time constant of the opposite sign to the control data, $\tau \approx -0.24$ s (leaky accumulation over only a quarter of a second). In contrast, the best-fitting lapse rate remained essentially unchanged from control. As described above, attempting to fit the inactivation data by keeping the time constant unchanged and increasing the lapse rate could qualitatively match the psychometric curve (magenta line, *Figure 4A*), but it produced an extremely poor fit relative to the full behavioral model (magenta cross in low likelihood region, *Figure 4B*). Thus, after bilateral FOF inactivation, the subjects behaved as if their lapse rate was unchanged, and their accumulator had become much leakier (*Table 1*).

To further validate these results we resampled the trials (with replacement) and refit the model on the resampled trials 300 times. Based on this analysis, only two parameters shifted significantly from the control data: $\lambda$ (95% C.I. = [−7.91 −1.31]; control value was 1.22) and Þ (95% C.I. = [0.10 1.50]; control value was 0.065). The change in $\lambda$ is both significant and substantial, and accounts for most of the change in performance. The change in decision boundary, Þ (pronounced 'sho') is significant but very small, corresponding a horizontal shift in the psychometric curve of only one click. It is largely due to the difficulty of performing a perfectly balanced bilateral infusion. In particular, one rat, A077, was

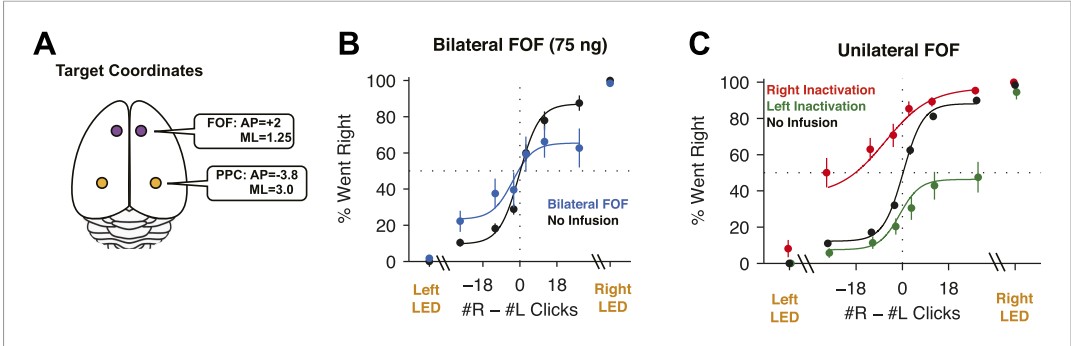

**Figure 3**. FOF Infusions. (**A**) Top-down view of rat cortex with the locations of the FOF and the PPC, into which cannulae were implanted. (**B**) Bilateral infusion of muscimol into the FOF results in a substantial impairment on accumulation trials but has no effect on side LED trials. In black are data from control sessions 1 day before an infusion (n = 8 sessions, 4 rats). In blue are data from bilateral FOF infusions (n = 8 sessions, 4 rats, 75 ng per side). The circles with error bars indicate the mean ± s.e. across sessions. Accumulation trials are binned by #R − #L clicks, spaced so there are equal number of trials in each bin. The lines are a 4-parameter sigmoid fit to the data. (**C**) Unilateral infusion of muscimol into the FOF results in a profound ipsilateral bias on accumulation trials but has no effect on side LED trials. In black are data from control sessions 1 day before an infusion (n = 34 sessions, 12 rats). In red are data from right FOF infusions (n = 17 sessions, 12 rats, 150 or 300 ng). In green are data from left FOF infusions (n = 17 sessions, 12 rats, 150 or 300 ng).

The following figure supplements are available for figure 3:

**Figure supplement 1**. Cannula coordinates and histology.

**Figure supplement 2**. Timeline of bias for each rat.

**Figure supplement 3**. FOF infusions cause profound impairment in the clicks task.

strongly biased during the bilateral FOF inactivations (*Figure 3—figure supplement 3A*). On average, the noise and lapse parameters also increased but due to large covariance between these parameters, it is not possible to say which of them was significantly shifted (*Figure 4—source data 1*).

To further examine the relative contributions of the $\lambda$ and Þ changes we fit two 1-parameter models. First, we fit the bilateral FOF data with a 1-parameter Þ model where only the decision boundary could change and the other 8 parameters were fixed at their best control data values. The log likelihood of the best Þ model was −1221.6, substantially worse than the model in which all 9 parameters were allowed to vary (−1102.5). Using Bayesian or Akaike Information Criteria (BIC or AIC; *Burnham and Anderson, 2004*), we find that the extra parameters are indeed justified (*Table 2*). Second, we fit the bilateral FOF data to a 1-parameter $\lambda$ model where only the accumulation time-constant could change. For this model, the best-fit value of $\lambda$ was −12.4 and the log likelihood of the model was −1121.1 compared to −1102.5 for the best 9-parameter model, a difference of only 18.5. According to BIC, the 1-parameter model is the more likely model, supporting the idea that the major effect of bilateral FOF inactivation was a change in the time constant of accumulation (*Table 2*).

To probe the conclusions derived from the trial-by-trial model fit, we used model-free analyses of the data (*Brunton et al., 2013*). Leaky accumulation with a time constant $\tau \approx -0.24$ s would result in short trials (with a stimulus duration less than a quarter of a second) being essentially unimpaired, while long duration trials would be more strongly impaired. This was indeed observed in the data, with a tight correspondence between the quantitative model and experimental data (chronometric curves, shown in *Figure 4C*). A further property of a leaky accumulator is that the more recent the clicks are, with respect to the end of the stimulus, the bigger their impact on the subject's decision. This was also observed in the data, again with a tight correspondence between quantitative model and experimental data (reverse correlation analysis, shown in *Figure 4D*).

These results indicate that the FOF is either (a) itself directly involved in the process of accumulating evidence, and the inactivations made the accumulator leaky; or (b) the FOF is a requisite

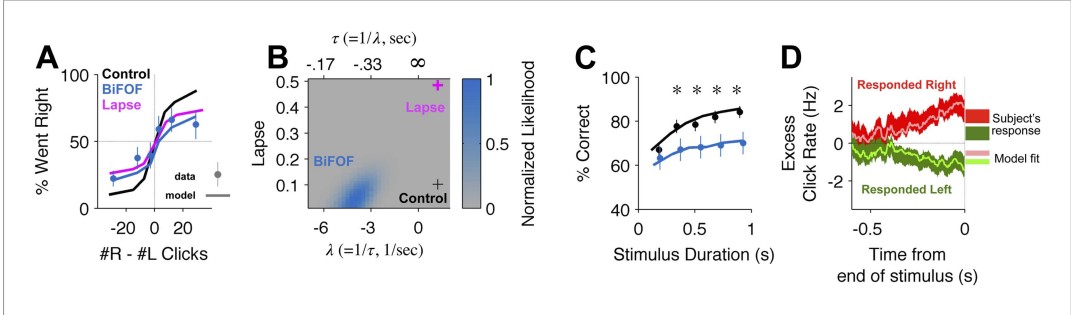

**Figure 4**. Bilateral FOF inactivation is best fit as a reduction in the time-constant of accumulation. (**A**) When analyzed in terms of the psychometric function, changes to either lapse rate alone or accumulation time constant alone can match the bilateral FOF inactivation data. The black line shows the psychometric curve from control data, collected 1 day before bilateral FOF sessions (n = 1526 trials). Blue dots with error bars show the experimental data from bilateral FOF inactivation sessions (n = 1809 trials). The magenta line is the psychometric curve obtained by fitting only the lapse rate parameter to the inactivation data, while keeping all other parameters at their control values (corresponds to magenta cross in panel **B**). The blue line shows the psychometric curve from the accumulator model fit to the inactivation data (corresponds to peak of blue likelihood surface in panel **B**), which has a change w.r.t. control in accumulation time constant $\tau$ (=$1/\lambda$), but no change in lapse rate. (**B**) Fitting the detailed click-by-click, trial-by-trial accumulator model (*Brunton et al., 2013*) to the inactivation data clearly distinguishes between lapse and $\lambda$. The panel shows the normalized likelihood surface, indicating quality of the model fit to the inactivation data as a function of the lapse and the $\lambda$ (=$1/\tau$) parameters. The black cross shows parameter values for the control data. The best fit to the inactivation data is at the peak of the blue likelihood surface ($\lambda = -4.15$, lapse = 0.048), significantly different from control for $\lambda$, but not different from control for lapse. This best-fit lambda corresponds to $\lambda = -0.241$ s, a substantially leaky integrator. (**C**) Performance as a function of stimulus duration for bilateral FOF sessions (blue, mean ± std. err.) and the control sessions 1 day before (black, mean ± std. err.). The lines are the chronometric curves generated by the accumulator model (for inactivation data, parameter values at peak of blue likelihood surface in panel **B**). (**D**) Reverse correlation showing the relative contribution of clicks from different times to the rats' decisions for data from bilateral FOF inactivation sessions. Compare to *Figure 2D*. The thick dark shading shows the mean ± std. err. across trials based on the rats' choices. The thin bright lines are the reverse correlation traces generated by the accumulator model (parameter values at peak of blue likelihood surface in panel **B**).

The following source data is available for figure 4:

**Source data 1**. MATLAB file containing resampled bilateral FOF model fits.

component in the output pathway of an accumulator with a long time constant, that is, the FOF is part of the chain of regions that transform evidence accumulated with a time-constant longer than 0.24 s into an orienting decision.

## Unilateral FOF inactivations produce a post-categorization bias

We also used an accumulator model to analyze the strong ipsilateral bias induced by unilateral FOF inactivations. The original model of Brunton et al. (*Figure 2—figure supplement 1*) contains only one parameter, the decision boundary Þ, that can generate a left/right bias. We therefore extended the model with three additional parameters, each of which represented a possible imperfection that could generate a side bias. Simultaneously fitting all 12 parameters substantially increased the computational difficulty of the fitting process. In particular, efficiently fitting the original model was made possible by analytical computation of the gradient (*Brunton et al., 2013*). Determining the gradient for 12 parameter model was outside the scope of this manuscript. We consequently took the strategy of first fitting the original 9-parameter model to the control data from 1 day before infusion sessions, and then, starting from those best-fitting parameter values, asking which of the bias-inducing single-parameter changes would best fit the data from the unilateral FOF inactivations (other parameters were held fixed at the control data best-fit values). In other words, we asked, 'if we changed only one parameter, which one would it be to best fit the data?' Finding the maximum likelihood value was made practical by the fact that each of the four fits performed was a single-parameter fit.

The four single-parameter changes we considered corresponded to hypotheses regarding possible functions of the FOF, and are conceptually illustrated in *Figure 5*. (a) First, we considered the

**Table 2.** Bilateral FOF model comparison

| Model | # of param. | Log likelihood | BIC | AIC |
|---|---|---|---|---|
| full model | 9 | −1102.5 | 2272.5 | 2223† |
| Þ model | 1 | −1221.6 | 2450.7 | 2445.2 |
| λ model | 1 | −1121.1 | 2249.7* | 2244.2 |

This table shows the three models fit to the bilateral FOF data (n = 1809 trials).
*indicates the model with the lowest (the most likely) Bayesian information criterion (BIC).
†indicates the model with the lowest (most informative) Akaike information criterion (AIC).
In this case, the AIC and BIC select different models, suggesting a better model may be somewhere in between.
That is, a model that includes the accumulator time-constant and perhaps a few additional parameters from the full model.

possibility that the FOF is part of the output pathway of the accumulator, perhaps part of computing or representing the animal's discrete choice after having categorized the accumulator value (into 'Go Right' vs 'Go Left' categories) (*Hanks et al., 2015*), potentially in the service of preparing a motor action (*Erlich et al., 2011*). Unilaterally perturbing the FOF might then bias this post-categorization representation. To implement this idea in the model, we added a parameter that biased outcomes after the R/L decision was made on each trial. Independently of the stimulus that led to the decision, we let a randomly-chosen fraction, $\kappa_R$, of right decisions and a randomly-chosen fraction $\kappa_L$ of left decisions be reversed (i.e., R → L and L → R; see *Figure 5A*). These reversals scale the vertical endpoints of the psychometric curve towards the Went Right = 50% level. The scaling is biased when $\kappa_R \neq \kappa_L$. (b) Next, we considered the possibility that since the FOF has been suggested as analogous to primate FEF, perturbing it might affect attentional processes, perhaps causing a lateralized sensory neglect that would bias the perceptual impact of auditory clicks from the two different sides. In other words, during unilateral inactivation, right and left clicks could have different magnitudes of their impact on the accumulating evidence ($C_R$ and $C_L$, instead of the single common $C$ of *Equation 1* in *Figure 2—figure supplement 1*). We described this as a 'unbalanced input gain' (see *Figure 5B*). (c) We next considered the possibility that the FOF plays a role in the accumulation process itself, and quantified biases in accumulation through an 'accumulation shift' that shifts the value of the accumulator, $a$, at the end of the stimulus (see *Figure 5C*, equivalent to Þ in *Figure 2—figure supplement 1*). Changes in this parameter will cause horizontal shifts in the psychometric curve. (d) In the fourth and final model, we considered a second possible form of lateralized sensory neglect, in the form of 'unbalanced input noise' (see *Figure 5D*). In this version of the model, right and left clicks could have different signal-to-noise ratios by having different values of the sensory noise parameter ($\sigma_{s,R}^2$ and $\sigma_{s,L}^2$, instead of the single common $\sigma_s^2$ of *Figure 2—figure supplement 1*). For each of models (a), (b), and (d), the original fit to the control data was constrained to be balanced. To fit the bias generated by unilateral FOF inactivation, that constraint was relaxed. For each model, we kept the corresponding ipsilateral parameter fixed, and let the contralateral parameter be free to best fit the data. The difference between the best-fitting contralateral parameter minus the ipsilateral parameter was then defined as the bias for that model.

Of these four models, the one that best fit the experimental data was (a), the post-categorization bias model in which the FOF is part of the output pathway of the accumulator (*Figure 6A*, *Figure 6—figure supplement 1B*). The value of $\kappa$ contralateral to the inactivation ($\kappa_C$) that best fit the data was 0.52, suggesting that on over 50% of trials rats reversed their contra-choices to ipsi-choices. The next best model (which was worse by ≈50 log-units than the post-decision model, *Figure 6A*) was (b), the biased input gain model. This model failed to accurately fit the data on difficult trials (in which |#Contra − #Ipsi Clicks| is small, *Figure 6—figure supplement 1C*). The best-fit model for (c), the accumulator shift model, was clearly a poor fit even when analyzed through psychometric curves, fitting particularly poorly for trials with a preponderance of contralateral clicks (*Figure 6—figure supplement 1D*). The worst fitting model was (d), the unbalanced input noise model (*Figure 6—figure supplement 1E*).

We verified that the post-categorization model was best by using a leave-one-session-out cross validation. For each of the 30 unilateral FOF infusion sessions, we fit the 4 models to the 29 other

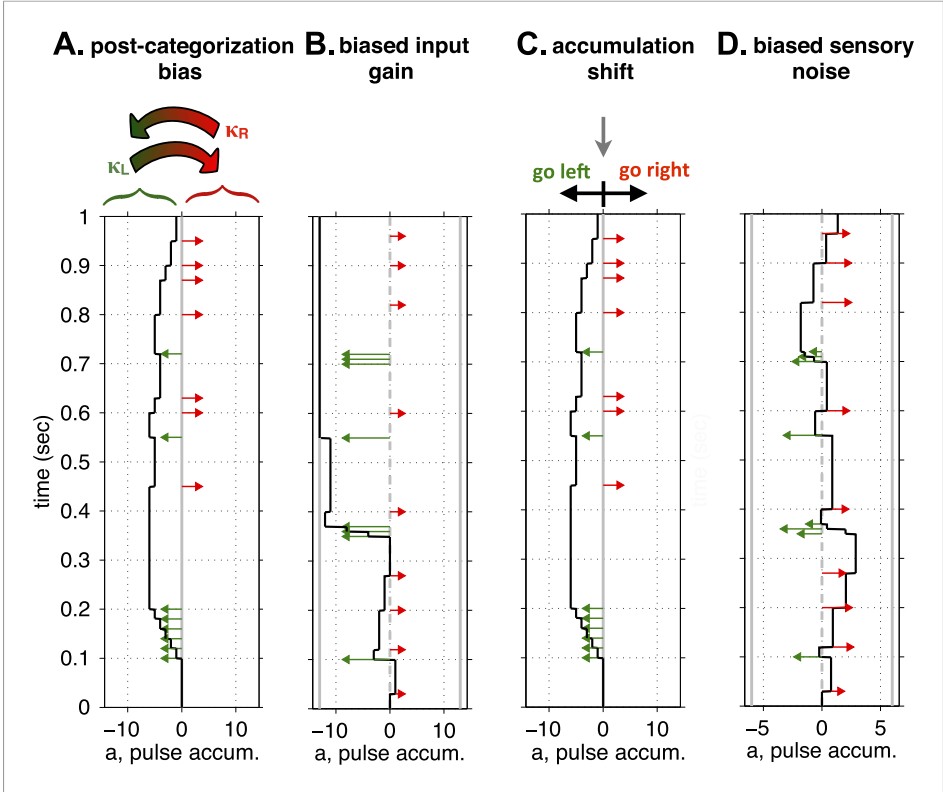

**Figure 5**. Conceptual illustration of four model parameters, used to quantify different sources of a lateralized bias.
(**A**) Post-categorization bias: after categorizing the accumulator value into 'Go Left' or 'Go Right' decisions,
a fraction, $\kappa_L$, of Left decisions are reversed into Right decisions, and a fraction, $\kappa_R$, are reversed from Right to Left.
(**B**) Biased input gain, which can be thought of as a form of sensory neglect: Left and Right clicks have different
impact magnitudes on the value of the accumulator. In this illustration left clicks have a much stronger impact, and
decisions will consequently be biased to the left. (**C**) Accumulation shift: before categorizing the accumulator into
'Go Left' vs 'Go Right' decisions (by comparing the accumulator's value to 0), a constant is added to the value of the
accumulator. (**D**) Biased sensory noise, which by differentially affecting signal-to-noise rations from the two sides,
can be thought of as a form of sensory neglect distinct from biased input gain: Left and Right clicks have different
magnitudes of noise in their impact. In this illustration, left clicks are more variable than right clicks, which biases
decisions to the right.

sessions and then evaluated the likelihood of the fit on the left-out session. Using this approach we
found a significant main effect of model on likelihood/trial (repeated-measures ANOVA $F(3,87) = 10.76$,
$p < 10^{-5}$) and Bonferroni-Holm corrected post-hoc $t$-tests reveal that all models were significantly
different from each other ($p < 0.005$) except the noise and accumulator shift models ($p > 0.4$).

To ask whether a combination of changes to two parameters could provide a significantly better
description of the data than our single-parameter fits, we estimated the 2-dimensional likelihood surface
for the two best models. This surface (*Figure 6B*) clearly demonstrated that fitting the data requires
a large shift away from the balanced control value in the post-categorization bias, but not in input gain.
Thus, the dominant effect of unilateral FOF inactivation could be parsimoniously explained by a post-
categorization bias, consistent with the known role of the FOF in movement planning (*Erlich et al., 2011*).
As described above, a feature of the post-categorization model is that the psychometric curve after
unilateral inactivations should be a vertical scaling of the control psychometric curve. This scaling was
found in the data, with a tight correspondence between the curve generated by the quantitative model
and the experimental psychometric curve (*Figure 6C*). A similar tight correspondence between model and
data were also found for the chronometric curves (*Figure 6D*) and the reverse correlation (*Figure 6E*).

Although computationally challenging, we have also explored whether higher dimensional models
might reveal a different set of results. Using the Metropolis–Hastings algorithm, we estimated the
best-fit posterior distribution for an 8-parameter model. This model contains the four bias parameters,

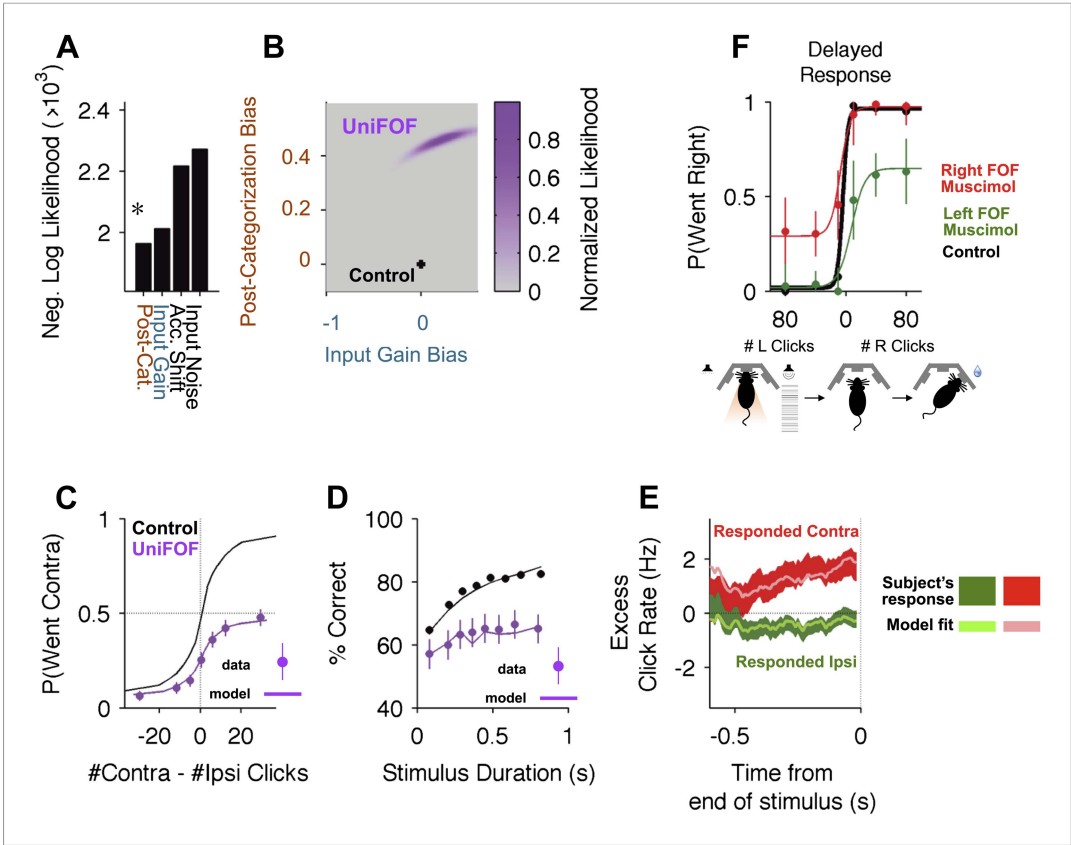

**Figure 6**. Unilateral FOF inactivation is best fit as a post-categorization bias. (**A**) A comparison of the likelihoods (i.e., best model fits) for the four different bias mechanisms illustrated in *Figure 5*. The post-categorization bias model is better than the next best model (biased input gain) by 50 log-units. (**B**) The 2-dimensional normalized likelihood surface for the two best single-parameter models: post-categorization bias and input gain bias. For visualization, we plot the contra-ipsi bias for the two parameters. That is, the difference between the contralateral and ipsilateral values for each parameter. The y-axis is $\kappa_C - \kappa_I$. The x-axis is $C_C - C_I$. By definition, in the control model (black marker) these biases are 0. The peak of the magenta likelihood surface for the inactivation data is significantly different from control for post-categorization bias (from 0 to 0.4588) but not significantly different from control for input gain bias. (**C**) Psychometric curves for control and inactivation data. The black line is the model fit to the control data (see *Figure 2A* for the data points). The magenta circles with error bars are experimental data from unilateral FOF inactivation sessions, and indicate fraction of Contra choice trials (mean ± binomial 95% conf. int.) across trial groups, with different groups having different #Contra − #Ipsi clicks. The magenta line is the psychometric curve generated by the post-categorization bias model. (**D**) Performance as a function of stimulus duration for data from control sessions 1 day before (black), and for data from unilateral FOF sessions (magenta, mean ± std. err.). The lines are the chronometric curves generated by the corresponding model (**E**) Reverse correlation analyses showing the relative contributions of clicks throughout the stimulus in the rats' decision process. The thick dark red and green lines are the means ± std. err. across trials for contralateral and ipsilateral trials. Thin light red and green lines are the reverse correlation traces generated by the post-categorization bias model. (**F**) Psychometric curves for single-sided trials in control (black), right FOF infusion (red) and left FOF infusion (green) sessions, demonstrate that even for very easy trials FOF infusions produce a vertical scaling, consistent with post-categorization bias.

The following figure supplements are available for figure 6:

**Figure supplement 1**. Psychometric and reverse correlation comparisons of data and model for unilateral FOF inactivations.

**Figure supplement 2**. Distribution of sample from 8-parameter model of unilateral FOF inactivation.

**Table 3**. Unilateral FOF model comparison

| Model | # of parameters | Log likelihood | BIC | AIC |
|---|---|---|---|---|
| Post-categorization bias | 1 | −1963.1 | 3934.4* | 3928.2† |
| Unbalanced input gain | 1 | −2013.1 | 4034.4 | 4028.2 |
| Accumulator shift | 1 | −2217.4 | 4443.0 | 4436.8 |
| Unbalanced input noise | 1 | −2272.7 | 4553.7 | 4547.4 |
| 8-parameter model | 8 | −1957.5 | 3981.1 | 3949.9 |

This table shows the three models fit to the unilateral FOF data (n = 3836 trials).
*indicates the model with the lowest Bayesian information criteria (BIC), that is, the most likely model.
†indicates the model with the lowest Akaike information criteria (AIC), that is, the most informative model.
The 1-parameter post-categorization model has the lowest AIC and BIC, supporting the view that the major effect of unilateral FOF inactivation is not related to the accumulation process per se.

as well as $\lambda$, $\sigma_a^2$, $\sigma_{S,I}^2$, and $\kappa_I$. For computational tractability, the remaining 4 parameters (initial noise, bounds, and the two click adaptation parameters) were fixed at the values of the best-fit control model. With 40,000 samples of this 8 dimensional distribution (*Figure 6—figure supplement 2*), we estimate that four parameters changed significantly from their control values. First, accumulator noise increased from $5 \times 10^{-4}$ to 0.746 (95% C.I. = [0.125 12.123], p = $10^{-4}$), which is still a small value and would have a negligible impact on behavioral performance. Second, accumulator shift changed from 0.065 to 0.323 ([0.136 0.890], p = 0.013), which would also have a minimal effect on overall reward rate. Third, $\kappa_I$, the changes from ipsi to contra choices decreased from 0.102 to 0.029 ([0.002 0.071], p = $10^{-4}$). Finally, supporting our earlier analysis, $\kappa_C$, increased from 0 to 0.498 ([0.176 0.536], p < $10^{-3}$) which results in a very substantial behavioral impact, since this sets the asymptotic performance on the easiest contralateral trials to ≈50%. As in the 2D model (*Figure 6B*) the gain of contralateral clicks increased (although not significantly). The likelihood of the best 8-parameter model was only marginally higher than the best 1-parameter post-categorization model (*Table 3*), consistent with the small shifts in the other parameters (log likelihood of the 8-D model: −1957.52; log likelihood of the 1-D model: −1963.08). Using AIC and BIC we find that the increase in likelihood is too small to justify the increase in number of parameters.

One of the characteristics of a post-categorization bias model is that since the biasing process occurs after the accumulated evidence has been categorized into 'Go Right' or 'Go Left', the bias is independent of whether trials are easy (large value of |#R − #L clicks|) or difficult (small value of |#R − #L clicks|). To further probe this hypothesis, we randomly intermixed regular accumulation trials with a new set of unusually easy 'single-sided' trials (*Figure 6F*). In these trials the speaker from only one side produced clicks at 100 clicks/s (noticeably higher than the 40 Hz rate on accumulation trials), lasting until the 'Go' signal indicating the end of center port fixation. Consistent with the post-categorization bias hypothesis, the unilateral FOF inactivations produced a vertically-scaled ipsilateral bias in these single-sided trials that was similar to that seen during the accumulation trials (*Figure 6F*, compare to *Figure 6C* data): that is, the bias was independent of how easy or how difficult the trials were.

## Unilateral PPC inactivations and comparison to unilateral FOF inactivations

Given that unilateral PPC lesions in rats lead to contralateral neglect (*Crowne et al., 1986*; *Reep et al., 2004*), that PPC has been posited as central to rodent perceptual decision-making (*Harvey et al., 2012*; *Carandini and Churchland, 2013*), and that neural correlates of the gradually accumulating evidence are found in PPC in our task (*Hanks et al., 2015*), we predicted that unilateral PPC inactivations would cause a strong contralateral impairment (or, in other words, in the context of our binary forced-choice task, an ipsilateral bias, similar to that seen with the FOF inactivations). Surprisingly, our unilateral PPC inactivations resulted in a small effect that was ≈10× smaller than the effect in the FOF. The average ipsilateral bias was 4.2 ± 2.4% (mean ± s.e.) (*Figure 7A*; *t*-test $t_{13}$ = 1.76, p > 0.1). Moreover, this small bias was largely due to data from the first infusion session in group 1 rats

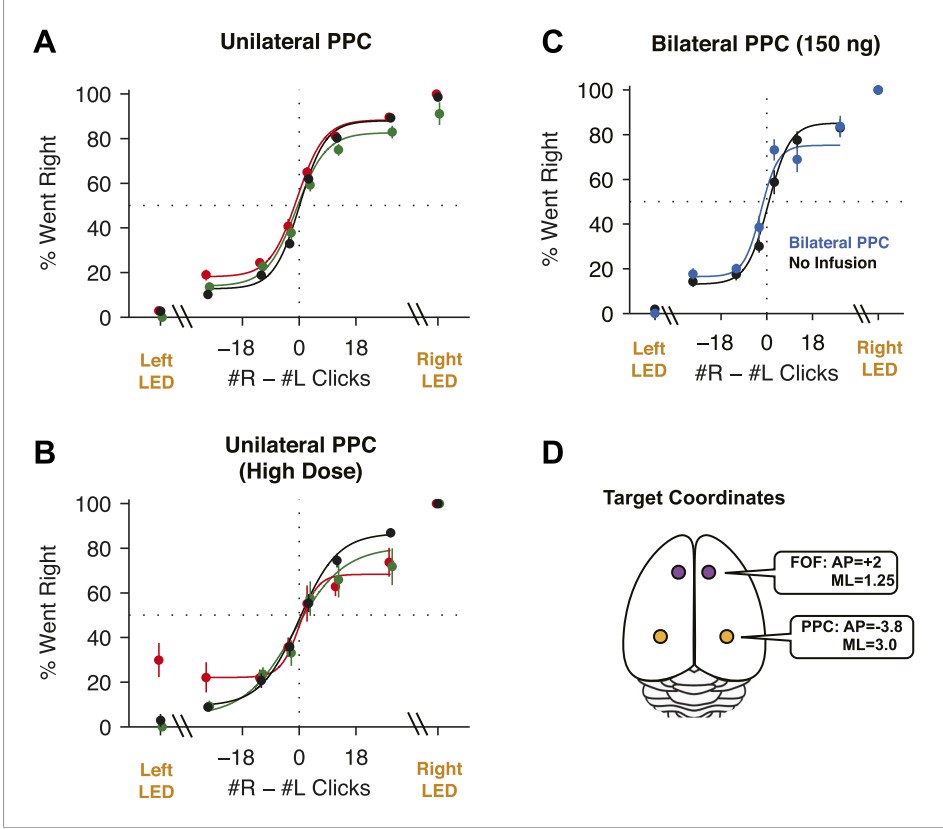

**Figure 7**. PPC Infusions. (**A**) As in *Figure 3B*, but for unilateral infusions of muscimol into the PPC, which result in a minimal impairment. In black are data from control sessions 1 day before an infusion (n = 65 sessions, 14 rats). In red are data from right PPC infusions (n = 31 sessions, 14 rats, 150 or 300 ng). In green are data from left PPC infusions (n = 34 sessions, 14 rats, 150 or 300 ng). (**B**) As in (**A**), but using very high doses of muscimol. Only very small effects are seen. In black are data from control sessions 1 day before an infusion (n = 11 sessions, 9 rats). In red are data from right PPC infusions (n = 3 sessions, 3 rats, 600 or 2500 ng muscimol). In green are data from left PPC infusions (n = 8 sessions, 8 rats, 600 or 2500 ng). (**C**) Bilateral infusion of muscimol into the PPC does not produce a markedly bigger impairment. In black are data from control sessions 1 day before an infusion (n = 8 sessions, 4 rats). In blue are data from bilateral PPC infusions (n = 8 sessions, 4 rats, 150 ng per side). (**D**) Schematic view of the brain, duplicated from *Figure 3A* to remind readers of the location of FOF and PPC on the cortical surface.

The following figure supplement is available for figure 7:

**Figure supplement 1**. PPC infusions have nominal effects on the Poisson Clicks task.

(*Figure 2—figure supplement 2A*) which led to a small but significant shift in rightward responding in right vs left infusions in the group 1 rats (p = 0.024, GLMM test). No consistent effects of unilateral PPC infusions were found in 10 subsequent infusion sessions with group 1 (a total of 11 group 1 unilateral PPC infusion sessions, *Figure 3—figure supplement 2*), even when the muscimol dose was substantially increased, to 600 ng (*Figure 7B*).

To test whether PPC inactivation could produce a quickly adapting effect (i.e., perhaps an effect from muscimol inactivation is observable only in the first session), we repeated our unilateral PPC inactivations using a second group of rats. However, no significant effect was found in any of the PPC infusion sessions in group 2 rats even on the first day of infusion (GLMM test, p = 0.92; *Figure 2—figure supplement 2B*). Notably, even extremely high doses (up to 2500 ng) of muscimol in PPC, were ineffective at biasing the rats on accumulation trials (*Figure 7B*). Thus, our data from group 2 suggest that the bias on the first day in group 1 occurred by chance.

Our PPC coordinates for group 1 and 2 (At 3.8 mm posterior and ≈3 mm lateral to Bregma) were based on the Paxinos and Watson rat atlas, the neural correlates of accumulation found at this location

(*Hanks et al., 2015*) and several published studies of rat PPC (*Paxinos and Watson, 2004*; *Nitz, 2006*; *Whitlock et al., 2012*). Nevertheless, some authors have suggested that PPC is slightly more posterior: 4–6 mm posterior to Bregma (*Kolb and Walkey, 1987*; *Wilber et al., 2014*). We therefore repeated the PPC experiments with a third group of rats, this time implanting cannulae at 4.5 mm posterior to Bregma (*Figure 3—figure supplement 1C*). Once again, as in group 2, there was no ipsilateral bias due to muscimol (at 300 ng) in the PPC (GLMM test, p = 0.47), nor was there a detectable effect on the first inactivation session.

Performance of side LED trials at regular muscimol doses (150 or 300 ng) was not significantly affected by PPC infusions (*Figure 7A*; *t*-test $t_7 = 1.0$, p > 0.35), and the response times on these trials were also unaffected (repeated-measures ANOVA, F(1,6) = 2.48, p > 0.15). At very large doses, a significant effect on side LED trials led to a small correlation between dose and bias for side LED trials (r = 0.43, p = 0.032; *Figure 8* yellow circles). Given the very large muscimol doses used, and the fact that visual cortical areas lie immediately posterior to PPC (*Paxinos and Watson, 2004*), this weak correlation on side LED trials may be a result of spread of muscimol to the adjacent visual cortex.

To summarize, unilateral inactivation of PPC did not reliably bias accumulation trials in three separately tested groups of rats. Based on the large doses used we suspected that the lack of effect was not due to a failure to inactivate PPC. There was no correlation between dose and bias magnitude in our PPC infusions on accumulation trials (*Figure 8*, yellow squares; p > 0.09), strongly contrasting with the significant correlation between dose of muscimol infused into the FOF and bias on accumulation trials (accumulation trials r = 0.85, p < $10^{-9}$; *Figure 8*, magenta squares). The correlation in the FOF data was specific to accumulation trials (p > 0.5 for side LED trials; *Figure 8*, magenta circles).

## Bilateral PPC inactivations

It is possible that during unilateral inactivations, the silenced PPC may be compensated for by the PPC of the opposite hemisphere. In this case, bilateral inactivations of the PPC should produce a behavioral impairment markedly larger than any small impairment found after unilateral inactivations. To probe this hypothesis, we initially used a high dose (300 ng per side) for bilateral PPC infusions in group 1 rats, but only 2 of 6 rats completed trials, with inconsistent results. The maximum dose at which subjects still performed substantial numbers of trials was 150 ng per side, double the dose per side for the bilateral FOF inactivations described above. During the bilateral PPC inactivation sessions there was a very small but significant 3.6% decrease in performance on accumulation trials (*Figure 7B*, p < 0.02 vs isoflurane, GLMM test). This effect was not individually significant in any rat (0/4). Critically, the effect size we found was not bigger–in fact, it was slightly smaller–than the average unilateral PPC effect, and thus does not provide support for the hypothesis of hemispheric compensation. Performance on side LED trials was not significantly different between bilateral PPC infusion and control sessions (*t*-test, p > 0.4; *Figure 7B*). Fitting the accumulator model to the bilateral PPC data, we find that the only parameter to change was the lapse (but the confidence intervals overlapped with the control value), suggesting that the effects on performance were unrelated to any specific aspect of the accumulation process.

## Free-choice trials

Our results from PPC contrasted with previous studies that found a strong effect of PPC inactivations in a mouse memory-guided navigation task (*Harvey et al., 2012*) and a strong effect of permanent unilateral PPC lesions in inducing contralateral neglect in rats (*Crowne et al., 1986*; *Reep et al., 2004*). This motivated us to seek a positive control task. The primate literature suggested an internally-guided decision task whose trials could be readily intermixed with our evidence accumulation task. *Wilke et al. (2012)* interspersed regular memory-guided saccade trials ('instructed' trials), in which a single saccade target was presented on each trial, with internally-guided 'free choice' trials, in which both an ipsilateral and a contralateral target were presented, and the monkey was rewarded regardless of its response choice. By design, subjects were free to respond as they pleased in free choice trials, and they typically displayed a bias towards one side or another in these trials. Wilke et al. found that muscimol inactivation of area LIP within PPC produced no effect on choices in the instructed memory-guided saccade trials, but produced a profound ipsilateral bias during intermixed free choice trials. Inspired by Wilke et al.'s results, we modified the task for seven of our group 2 and group 3 cannulated rats (The six group 1 rats and one group 3 rat had been already sacrificed for histology). We randomly intermixed 25% free choice trials with 65% accumulation trials and 10% Side LED trials (*Figure 9A*). Free-choice trials were indicated by a lack of auditory click stimuli, and by illumination of both side LEDs after the animals had withdrawn

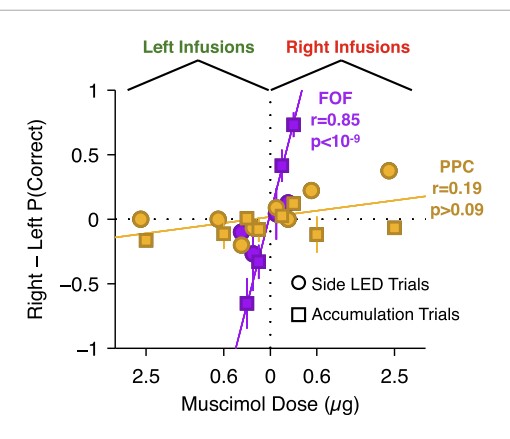

**Figure 8**. Summary of dose-bias relationship for all unilateral infusions. Infusions into the FOF are in magenta and into the PPC are in yellow. Circles indicate bias on LED trials, squares indicate bias on accumulation trials. The magenta line is the linear fit between signed dose of FOF infusion (+for right infusions, −for left infusions) and performance bias (right − left % correct) on accumulation trials (r = 0.85, p < 10⁻⁹). The yellow line is the linear fit between signed dose of PPC infusion and performance bias on accumulation trials (r = 0.19, p > 0.05). The plotted x-location of the side LED trials are slightly offset for visualization.

from the center port. We refer to sessions with interleaved accumulation, side LED, and free-choice trials as 'free-choice' sessions. After a few free-choice sessions with no infusions, rats performed the mix of trials reliably, and expressed a consistent bias on free choice trials but no detectable bias on accumulation trials.

In remarkable parallel to Wilke et al.'s results in primates, unilateral PPC inactivations (300 ng of muscimol) during free-choice sessions produced a very strong and reliable ipsilateral bias on free choice trials (*Figure 9B,C*; t-test $t_{26}$ = 3.70, p = 0.001). The strong ipsilateral bias in free choice trials was observed even while, consistent with our previous PPC inactivations, there was no ipsilateral bias on the intermixed accumulation trials (t-test $t_{26}$ = −0.99, p = 0.329) nor on the Side LED trials (t-test $t_8$ = 1.42, p = 0.194; *Figure 9C*). The free-choice bias was highly reproducible: 85% (23/27, sign-test, p < 0.001) individual rat PPC inactivation sessions produced an increased fraction of ipsilateral free choices when compared to free choices on immediately preceding control days (see *Figure 9B* for an example of PPC infusions that were selected to 'push' the rats away from their innate preference seen on the day before and the day after the infusion). The effect on free choice trials was thus similar in its robustness and reproducibility to the effect of unilateral FOF inactivation on accumulation trials. These free choice trial inactivation results provide a clear positive control for our PPC inactivations. Moreover, they are consistent with the parietal neglect literature in both rats (*Crowne et al., 1986*; *Reep et al., 2004*) and primates (*Mesulam, 1999*).

Inactivation of FOF, like the PPC, also induced an ipsilateral bias on free choice trials (t-test, $t_{24}$ = 3.86, p = 0.001). Consistent with our previous experiments, FOF inactivations in free-choice sessions continued to produce an ipsilateral bias on accumulation trials (t-test $t_{24}$ = 4.85, p < 0.001) but not on side LED trials (t-test $t_{18}$ = 1.65, p = 0.117; *Figure 9D*).

## Simultaneous FOF and PPC inactivation

Are there any conditions under which silencing the PPC could affect choices in auditory click accumulation trials? To probe whether inactivation of the FOF could reveal an effect of PPC inactivation, we bilaterally inactivated the FOF while simultaneously infusing 300 ng of muscimol unilaterally into the PPC. This combination of infusions produced a significant 15.1% bias ipsilateral to the side of the PPC infusion (*Figure 9E*, p < 0.0012, GLMM test). These data constitute a second positive control for our unilateral PPC inactivations. The data furthermore suggest that during auditory evidence accumulation the PPC may have a real but weak influence on choice that is normally overridden by a stronger signal from the FOF.

## Discussion

In two-alternative forced choice tasks driven by accumulation of sensory evidence, such as the random dots task used with primates (*Newsome et al., 1989*) or the Poisson Clicks task used here with rats (*Brunton et al., 2013*), subjects gradually accumulate evidence over time; make a decision by categorizing the graded value of the accumulated evidence into a binary choice; use their decision to prepare a movement; and finally execute their decision-reporting motor act (these different components could potentially overlap). In primates, five recurrently interconnected brain regions have been associated with the overall process (superior colliculus, striatum, PPC, FEF, and dlPFC; *Horwitz and Newsome, 1999*; *Kim and Shadlen, 1999*; *Shadlen and Newsome, 2001*;

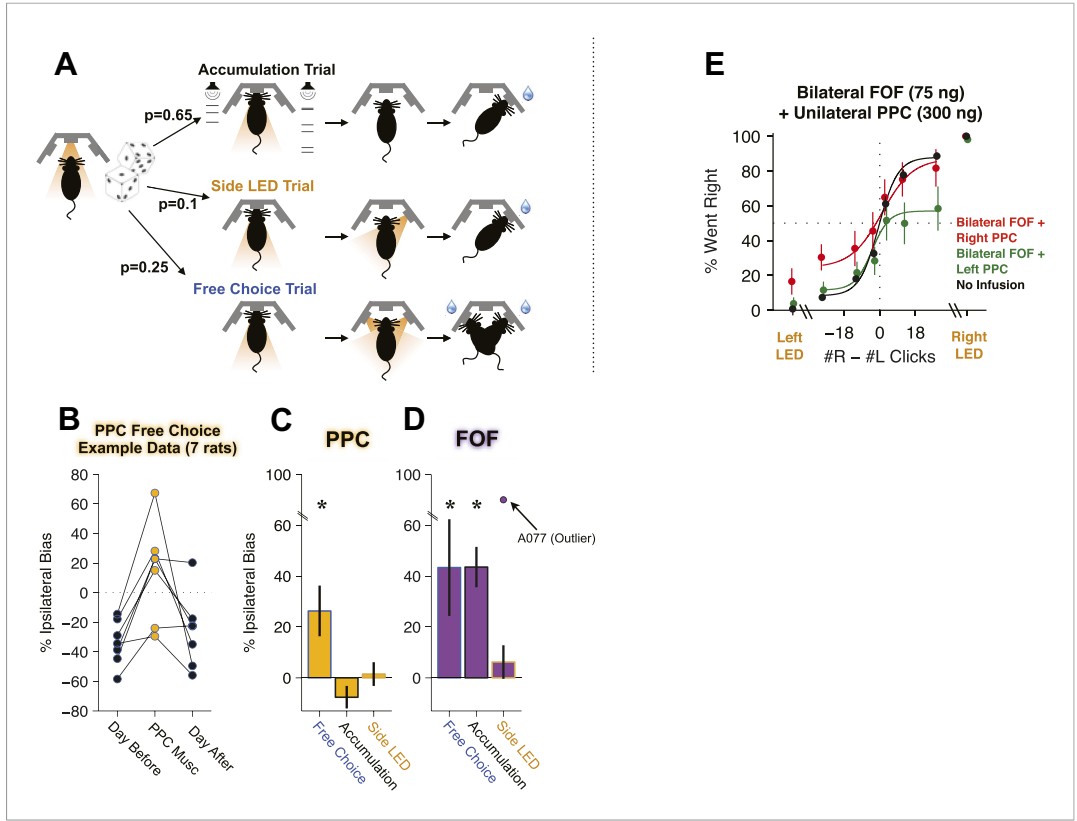

**Figure 9**. Unilateral PPC inactivations induce a strong ipsilateral bias during internally guided decisions, and can induce a strong bias on accumulation trials if the FOF is bilaterally inactivated. (**A**–**D**) Effect of unilateral PPC or FOF inactivations on free choice trials intermixed with regular accumulation trials. (**A**) A schematic of the three interleaved trial types: Accumulation, Side LED and Free Choice trials. Accumulation and side LED trials proceeded as described in *Figure 1*. On free choice trials (25% of all trials), after withdrawal from the center port both side LEDs were illuminated and rats were rewarded for going to either port. (**B**) After a few sessions, rats quickly developed an intrinsic bias on free choice trials even while showing no bias on instructed trials (Accumulation and side LED trials). When muscimol was infused into the PPC free choices were significantly biased toward the side of the infusion. This panel shows an example from a single infusion day where the side of the infusions was chosen to be opposite to their intrinsic free-choice bias, together with data from the previous and subsequent control day. (**C**) Unilateral PPC infusions generated a significant 26 ± 9% (mean ± s.e. across rats, n = 7) ipsilateral bias on free choice trials compared to control sessions (the day earlier). During infusion sessions there was a small 8 ± 4% (mean ± s.e., n = 7 rats) contralateral bias on accumulation trials, perhaps compensatory to the free choice ipsilateral bias. There was no effect on side LED trials (**D**). Unilateral FOF infusions generated significant ipsilateral biases on free choice and accumulation trials, but not on side LED trials. One rat (A077) was extremely biased (90% Ipsilateral choices) on Side LED trials during FOF inactivation, indicated as an outlier. *p < 0.01. (n = 25 session, 7 rats) (**E**) Combined bilateral infusion of muscimol into the FOF with unilateral infusion of muscimol into the PPC results in a substantial ipsilateral bias on accumulation trials. In black are data from control sessions 1 day before an infusion (n = 32 sessions, 4 rats). In red are data from bilateral FOF infusions with right PPC infusion (n = 8 sessions, 4 rats, 75 ng per side in the FOF, 300 ng in right PPC). In green are data from bilateral FOF infusions with left PPC infusion (n = 8 sessions, 4 rats, 75 ng per side in the FOF, 300 ng in left PPC). See *Figure 9—figure supplement 1* for the individual rat results and GLMM fits.

The following figure supplement is available for figure 9:

**Figure supplement 1**. Simultaneous infusion data for each rat.

*Roitman and Shadlen, 2002*; *Ratcliff et al., 2007*; *Ding and Gold, 2010*, *2012*; *Heitz and Schall, 2012*; *Mante et al., 2013*), but despite some theoretical suggestions (*Lo and Wang, 2006*), the specific contributions of each brain region to the different aspects of the overall process, and the circuit logic of the network, remain unclear. Here we focused on the role of two rat cortical areas, posterior parietal cortex (PPC) and frontal orienting fields (FOF), that are considered critical for rodent decision-making

(*Erlich et al., 2011*; *Sul et al., 2011*; *Harvey et al., 2012*; *Carandini and Churchland, 2013*), and that display neural correlates of gradually accumulating auditory evidence (*Hanks et al., 2015*). As in primates, the specific roles of each of these rat areas within the overall evidence accumulation and decision process remain undetermined.

Using a within-subject design, we implanted cannulae in both of these areas, and carried out pharmacological inactivations, quantifying the impairments by fitting to the data detailed models of the decision-making process for the Poisson Clicks task. Different parameters of the models quantified different possible variations from control behavior. We also compared inactivation effects on the Poisson Clicks task ('accumulation trials') to three types of control trials: free-choice (in which the animal was free to choose either of two visual stimuli to obtain a reward); single-sided (in which decisions were guided by a simple auditory stimulus that did not require gradual evidence accumulation, and rats had to withhold their response for several hundred milliseconds after receiving enough information to make their decision); and side LED trials (in which decisions were guided by a simple visual stimulus that did not require gradual evidence accumulation, and rats were free to report their decision as soon as they made it).

In other studies with the same behavioral task, we have used optogenetic inactivation (*Hanks et al., 2015*). Although optogenetics allows high temporal resolution inactivation, its radius of effect in our hands is only ≈750 μm (see Extended Data *Figure 7* of *Hanks et al., 2015*). In contrast, inactivating larger regions is much more readily achieved with muscimol, for which the radius of inactivation can be increased simply by increasing the infusion dose. In addition, muscimol directly inactivates all neurons within its radius of effect, not only infected neurons. Moreover, in some tasks, including the Poisson Clicks task, we find that the behavioral effect of optogenetic silencing begins to decay after a few weeks, while the effect of muscimol is stable. This stability in particular was essential for the current study, which used within-subject manipulations over hundreds of days. Optogenetic and pharmacological silencing therefore have complementary advantages and disadvantages. Here we focused on pharmacological inactivation.

## Role of FOF

Our results demonstrate that the FOF is an essential part of the circuit for decisions driven by accumulating evidence (*Figure 3*). Unilateral FOF inactivations had a strong effect on both accumulation trials, signaled by auditory clicks, and on free choice trials, signaled by a bilateral visual stimulus (*Figure 9*), indicating that the effects are not specific to a single sensory modality. Critically, the model-based analyses suggested a specific location for the FOF within the functional process chain required by our accumulation of evidence task. The specific suggestion is that the FOF is not part of the accumulator but is instead part of the premotor output pathway that leads from the graded evidence accumulator to the decision-reporting motor act. This suggestion is derived from (a) the sharp reduction in the accumulation time constant induced by bilateral FOF inactivations (from slightly unstable $\tau \approx +0.8$, to very leaky $\tau \approx -0.24$ s), which demonstrates that the FOF is either involved in the accumulation process itself, or is part of the output pathway of the accumulator (*Figure 4*); (b) the effects induced by unilateral inactivation of the FOF, which are captured in quantitative detail, for both accumulation trials and single-side trials, as having induced a post-categorization bias that is subsequent to the accumulation process (*Figure 6*); and (c) the lack of an effect of FOF inactivations on side LED trials, which rules out a simple motor role for the FOF (*Figure 3*). A parsimonious explanation of this set of results is thus that the FOF is a requisite premotor component of the output pathway of an evidence accumulator with a long time constant (>0.24 s). This suggestion is different from, but consistent with, the FOF's known role in short-term memory, as well as the FOF's greater importance in memory-guided (i.e., long time constant) vs sensory-guided (short time constant) orienting decisions (*Erlich et al., 2011*). In decisions driven by gradual accumulation of evidence, the gradual accumulation process occurs prior to the binary decision. An intriguing possibility suggested by our results that the decision process itself–that is, the categorization of the gradually accumulated evidence into a binary choice–might be performed in the FOF, perhaps in conjunction with the superior colliculus (*Lo and Wang, 2006*).

It is possible that unilateral FOF inactivations would induce a hemispheric imbalance so strong that a real but nuanced role for the FOF in gradual accumulation might have been obscured by a strong post-categorization bias. We nevertheless currently favor the interpretation of the FOF's role as subsequent to, not part of, the graded accumulator. We favor this interpretation first, because of its parsimony; second, because parallel electrophysiological work from our laboratory found that the

representation of the accumulated evidence in the FOF could be approximately described as the answer to the categorical question 'if the GO signal came now, which side port should I choose?'; third, using AAV-CaMKII-eNpHR3.0 for optogenetic inactivation we found that transient unilateral silencing of the FOF was unable to cause an effect on behavior if it occurred during the evidence accumulation period, sufficiently prior to the GO signal (*Hanks et al., 2015*).

Those results, in combination with the pharmacological data, three control trial types, and model-based analyses reported here, all consistently support the interpretation of the FOF as a requisite component of the output pathway of an evidence accumulator with a long time constant (>0.24 s). This view is consistent with the fact that side LED trials, which involve decisions that do not require gradual evidence accumulation or storing a motor plan in short-term memory, are not impacted by FOF inactivations. Such decisions, including perhaps decisions that require auditory evidence accumulation over only short times (<0.24 s), may depend on pathways that bypass the FOF, perhaps involving direct connections from auditory cortex to the striatum (*Znamenskiy and Zador, 2013*).

With pharmacological infusions, one concern is spread of inactivation to other structures. We based the volume and concentration of our infusions on previous literature to achieve inactivation volumes of 1 mm radius for our small doses and 3 mm radius for our largest doses (*Martin, 1991*; *Krupa et al., 1999*). This spread is within the anterior-posterior bounds of FOF, but could have spread medially to cingulate cortex (CG1) or laterally to M1 (*Paxinos and Watson, 2004*). The white matter below FOF prevent spread ventrally into the basal ganglia. In a previous study, we directly tested the effects of M1 inactivation and found them to be weaker than FOF inactivations and also they affected sensory-guided and single-sided trials equally (*Erlich et al., 2011*). As such, it is unlikely that the effects we are attributing to FOF are due to M1 inactivation. We targeted our inactivations to 2 mm anterior and 1.25 mm lateral to Bregma. According to the Rat Atlas (*Paxinos and Watson, 2004*) CG1 may be within the spread of the drug. However, according to a recent cell-based mapping technique the medial boundary of FOF has been underestimated (*Brecht et al., 2004*), suggesting that most of spread of drug would be within FOF. Moreover, the CG1 is thought to play a role in cost-benefit decisions (*Hillman and Bilkey, 2012*; *Holec et al., 2014*), cognitive flexibility (*Ragozzino and Rozman, 2007*), or emotional reactivity (*Bissière et al., 2008*), not in movement planning. Therefore, the effects observed are more likely due to changes in FOF rather than an adjacent cortical region.

## Role of PPC

Given the view that the PPC plays a key causal role in rodent perceptual decisions (*Harvey et al., 2012*; *Carandini and Churchland, 2013*) and that neural activity in PPC displays correlates of accumulating auditory evidence (*Hanks et al., 2015*), we were surprised to find that unilateral inactivation of the PPC did not cause a side bias on accumulation, side LED, or single-sided trials. Compensation from the unperturbed hemisphere did not explain the lack of an effect, because bilateral inactivation of the PPC caused a minimal decrease in performance (3.6%) that was not significantly different from unilateral inactivations. Thus, the PPC seems to play a far more subtle role than the FOF in choice behavior during decisions guided by auditory evidence accumulation.

Unilateral PPC inactivations did cause a strong ipsilateral bias under two conditions: in internally-guided decisions (free choice trials, signaled by a visual stimulus, *Figure 9A–C*), and when the FOF was simultaneously bilaterally inactivated in accumulation trials (auditory stimulus, *Figure 9E*). These results suggest that in the intact brain, weak but real side choice signals from the PPC may be overridden by stronger signals from the FOF. They also suggest that the PPC's role may not be specific to a particular sensory modality.

The PPC is relatively extended in the medial-lateral direction (≈4 mm) but it is thin (≈0.75 mm) in the anterior-posterior direction (*Paxinos and Watson, 2004*; but see *Wilber et al., 2014*). The anatomical characteristics of the PPC pose two possible confounds: insufficient inactivation of the PPC in the medial-lateral direction and potential spread to adjacent regions in the anterior-posterior direction. If we assume the spread of muscimol across the cortical is isotropic (white matter acts as a physical barrier, so we assume that the ventral spread is restricted), then an individual injection, due to the wide and thin shape of PPC, will either fail to inactivate all of PPC or spread to adjacent regions. Based on the literature (*Martin, 1991*; *Krupa et al., 1999*) we expect our smallest infusion to have a ≈1 mm radius and our largest to have a >3 mm radius. This technical limitation would have posed a serious challenge for our results if we observed very different effects of our small and large infusions. However, despite using a wide range of doses, we observed no significant dose–response effect with unilateral PPC

infusions (*Figure 8*). A clear dose–response relationship (as we found in the FOF) would indicate that bigger infusions were silencing more and more PPC neurons required for behavior in the task. Instead, the fraction of inactivated PPC neurons had no impact on the magnitude of the behavioral effect, which is what we would expect if PPC neurons were not required for choice behavior in the task. Nonetheless, further experiments would be required to completely rule out the possibility that a small number of PPC neurons, in the most medial or lateral edge of the PPC, were spared and that these few neurons were sufficient to support intact behavioral performance on the Clicks task.

Spread of muscimol into areas immediately anterior or posterior to the PPC was inevitable with our largest doses, and, in the two cases where positive effects were observed, this raises concerns about region specificity. Immediately anterior to the PPC is the somatosensory cortex (for the trunk). Particularly in light of the intact motor capacities, as indicated by the intact side LED trials, the somatosensory cortex is not expected to have caused any of the observed effects. Immediately posterior to the PPC are a set of individually small visual areas, collectively referred to as secondary visual cortex (V2; *Coogan and Burkhalter, 1993*; *Montero, 1993*; *Wang and Burkhalter, 2007*). These visual areas are unlikely to be involved in auditory click accumulation trials, and are therefore not expected to have caused the effects we saw after simultaneous unilateral PPC and bilateral FOF inactivations (*Figure 9E*). However, the bias we observed on free-choice trials may have been partly due to an effect in one or more of these small secondary visual areas. Nevertheless, ipsilateral biases in untrained orienting responses (potentially analogous to the free-choice task) due to PPC lesions have been observed in the rat even when those choices were to tactile or auditory stimuli (*Corwin et al., 1996*; *Burcham et al., 1997*, *1998*). We also note that our free choice results closely parallel the results from primate PPC free choice experiments (*Wilke et al., 2012*), which do not suffer from this spillover concern.

A minimal effect following PPC inactivation in our rat auditory click accumulation task is reminiscent of Guo et al.'s, finding of no effect after PPC inactivation in a mouse somatosensory-cued, memory-guided task (*Guo et al., 2014*). But it contrasts with Harvey et al.'s finding of a severe performance impairment after PPC inactivation in a mouse visually-cued, memory-guided navigation task (*Harvey et al., 2012*). Following our own preliminary reports of PPC inactivations in the Poisson Clicks task (*Erlich et al., 2012*, *2014*), *Raposo et al. (2014)* reported an impairment after rat PPC inactivations in a visual, but not in an auditory, version of a closely related task. This contrasts with a preliminary report from Yates et al. that suggested no effect from primate PPC inactivation in a visual accumulation of evidence task (*Yates et al., 2014*), as well as with multiple reports of no effect from primate PPC inactivation in visual memory-guided saccade tasks (*Chafee and Goldman-Rakic, 2000*; *Wardak et al., 2002*; *Liu et al., 2010*; *Wilke et al., 2012*). The Raposo et al. results therefore suggest that rodent PPC may be unlike primate PPC in being required for accumulation of evidence for a specific modality, vision. Nevertheless, the precise location of the border between rodent PPC and visual cortex, as well as the amount of inter-animal variability in this location, remain active research questions (*Wilber et al., 2014*; see, for example, *Reep et al., 1994*; *Corwin and Reep, 1998* for definitions from the same authors, based on the same data, that alternately describe the border as located at −5.0 mm from Bregma or −4.4 mm from Bregma). The uncertainty in the location of the PPC/visual cortex border raises an alternative possibility, which is that the impairments in memory-guided visual tasks observed by Harvey et al. and Raposo et al. (and potentially the free-choice effects seen here) could have been due to inactivation spillover into one of the immediately adjacent small secondary visual areas, as described above, rather than inactivation of the PPC itself. This would make all the inactivation results (*Harvey et al., 2012*; *Guo et al., 2014*; *Raposo et al., 2014*; *Yates et al., 2014*; and the results presented here.), including mouse, rat, and primate, fully consistent with each other. While such a reconciliation of results across multiple species may have some intellectual appeal, we emphasize that experiments that would either clearly support or rule out this possibility remain to be done.

## Comparison to primates

To date, the accumulation of evidence literature is mostly composed of electrophysiological experiments with primates. Based on a number of criteria, the rat PPC and FOF have been suggested as analogous to the primate PPC and FEF, respectively (*Kolb and Walkey, 1987*; *Reep and Corwin, 2009*; *Erlich et al., 2011*), and accumulation of evidence signals very similar to those in primates have been found in these two rat areas (*Hanks et al., 2015*). It is therefore tempting to speculate that our results in rat might also hold in primate, and to consider potential interpretations that would be consistent across mammalian model systems.

There have been no published inactivation experiments in primate PPC or FEF during accumulation of evidence tasks. There have, however, been inactivation experiments in related tasks, which in general are in agreement with our findings: prefrontal perturbations strongly bias behavior while posterior parietal perturbations do not. In memory-guided saccade tasks, LIP inactivations have no effect on the choice of saccading either towards or away from the correct hemifield (*Chafee and Goldman-Rakic, 2000*; *Wardak et al., 2002*; *Liu et al., 2010*; *Wilke et al., 2012*), while FEF and prefrontal inactivations reliably generate profound ipsilateral choice biases (*Sommer and Tehovnik, 1997*; *Dias and Segraves, 1999*; *Chafee and Goldman-Rakic, 2000*). Similarly, in a covert visual search task, LIP inactivation has no effect on error rates (*Wardak et al., 2002*), while inactivation of the FEF generates significant increase in error rates for contralateral targets (*Wardak et al., 2006*). Again similarly, in a memory-guided task with distractors, *Suzuki and Gottlieb, (2013)* found no errors after LIP inactivations while finding significant contralateral errors after prefrontal cortex inactivations.

We are aware of only a few studies where LIP inactivation produces choice biases (*Wardak et al., 2002*; *Balan and Gottlieb, 2009*; *Wilke et al., 2012*). One of these studies (*Wilke et al., 2012*) inspired our intermixing free choice trials with accumulation trials. Consistent with a good analogy between rat PPC and primate PPC, our results in rats closely paralleled Wilke et al.'s results in primates, with our accumulation of evidence trials playing the role of their memory-guided saccade trials (*Figure 8*). Also consistent with our rat data and with a good analogy between rat and primate PPC, a preliminary report has suggested that unilateral primate PPC inactivations have no effect on accumulation of evidence trials, while causing an ipsilateral bias on free choice trials (*Yates et al., 2014*).

There has been one perturbation study in the primate FEF during an accumulation of evidence task (*Gold and Shadlen, 2000*). This microstimulation study concluded that 'developing oculomotor commands may reflect the formation of the monkey's direction judgement' (i.e., its decision). Our results in rat FOF are consistent with those of Gold and Shadlen, but go considerably further in specifically suggesting the FOF as a requisite premotor component of the output pathway of a long time constant (>0.24 s) evidence accumulator.

There has been one perturbation study in the primate PPC during an accumulation of evidence task (*Hanks et al., 2006*). This microstimulation study used a reaction time version of the Random Dots task, and found a pattern of results following LIP microstimulation that could be quantitatively explained in an accumulation-to-bound model if the microstimulation added a small constant offset to the value of the accumulator (*Hanks et al., 2006*). Unlike the task used by Hanks et al., our task (and that of *Yates et al., 2014*) was not a reaction time task, which could explain the difference in results. An alternative possibility, noted in their discussion (*Hanks et al., 2006*), and which would reconcile our results with theirs, is that microstimulation may activate axon terminals or fibers of passage (*Histed et al., 2009*, *2013*; but see *Tehovnik and Slocum, 2013*). The behavioral effects produced by microstimulation of LIP may thus have been due to activation of neurons with somata not in LIP, but in regions of the brain that project to LIP. Since muscimol does not affect axons or fibers of passage, this possibility would be consistent with our data.

If the PPC plays a causal role in internally-guided decisions, but does not play a causal role in choice behavior during accumulation of evidence tasks, what role is then played by the firing rate correlates of evidence accumulation signals observed in PPC, in both rats and primates? Accumulation signals are also correlated with confidence (*Kiani and Shadlen, 2009*; *Komura et al., 2013*). One possibility is that, instead of being used to drive choice behavior, the accumulation signals observed in the PPC are used for computing choice confidence (*Kiani and Shadlen, 2009*). Confidence could be part of a process for optimizing behavior over many trials, or, in a reaction time version of the task, confidence could be part of determining when the subject chooses to commit to a decision.

## Conclusion

In the rat, our data now suggests a specific functional role for the FOF in decisions driven by accumulation of evidence: the FOF appears to be a requisite component of the output pathway of an evidence accumulator with a long time constant (>0.24 s). Decisions with shorter processing times may involve circuits that bypass the FOF. It is possible that categorizing the graded accumulator's value into a discrete choice–the final decision itself–could occur in the FOF. In contrast, the PPC seems to play a surprisingly minimal, or subtle, role in choice behavior during decisions guided by accumulation of evidence, even while it plays an important role in internally-guided decisions. Neither region appears likely to play a major causal role in the gradual evidence accumulation process per se.

# Materials and methods

## Subjects

Animal use procedures were approved by the Princeton University Institutional Animal Care and Use Committee and carried out in accordance with National Institutes of Health standards. All subjects were male Long-Evans rats (Taconic, NY). Rats were placed on a restricted water schedule to motivate them to work for water reward. Rats were kept on a reversed 12 hr light–dark cycle and were trained in their dark cycle.

## Behavior

We trained 14 male Long-Evans rats on the Poisson Clicks accumulation task (*Figure 1*). Training took place in a behavior box with three nose ports (left, center and right), and with two speakers, placed above the right and left nose ports. Each accumulation trial began with a visible light-emitting diode (LED) turning on in the center port. This cued the trained rat to place its nose in the center port, and keep it there until the LED was turned off. We refer to this period as the 'nose in center' or 'fixation' period. The duration of fixation was 2 s for all accumulation trials. During the fixation period a variable duration auditory stimulus (0.1–1 s, experimenter controlled) would play, consisting of two randomly timed trains of clicks, playing simultaneously, one from the left and one from the right speaker. At the end of the auditory stimulus, the LED in the center port would extinguish, which was the signal to the rat to make their response by poking into one of the side ports. The timing of the clicks from each speaker was generated by two independent Poisson processes. The total generative click rate (left + right rate) was held constant at 40 clicks/s, and trial difficulty was controlled by adjusting the relative left vs right rates, as well as the duration of the stimulus. Trials where rats exited the center port during the fixation period were considered violation trials, aborted, and a new trial was started. These trials are not included in any analyses.

To test whether the apparent vertical scaling of the psychometric curve after unilateral FOF inactivations was due to a lack of asymptotically easy trials, we interleaved 'single-sided' trials with accumulation trials (and side LED trials in some sessions). A single-sided trial was much like an accumulation trial, but all the clicks came from one speaker and were played at a Poisson rate of 100 Hz. Since the 100 Hz stimuli were easily distinguished from 40 Hz stimuli and these trials did not require accumulation (all clicks on one side), rats probably made their decision quickly. However, they were still required to wait until the go cue. These trials comprised ≈8% of a trials in a session.

In order to control for motor effects of inactivations, in group 2 and 3 rats, we included, randomly interleaved with other trial-types, 'side LED' trials. On side LED trials no sounds were played during fixation, which lasted 1 s. Immediately after the end of fixation, one of the two side ports was illuminated, indicating availability of reward at the lit port (*Figure 1*). The right and left side LED trials, together, comprised ≈10% of the total trials.

In order to find a task that was sensitive to inactivation of the PPC, we randomly interleaved 'free-choice' trials with the other trial-types. Free-choice trials were similar to side LED trials except at the end of fixation both side LEDs were illuminated and rats were rewarded regardless of whether they poked in the right or left nose port. These sessions took place after all of the experiments presented in *Figures 3–7, 8E*.

Control non-infusion sessions (used to generate *Figure 2—figure supplement 2*) with poor performance (<70% correct overall or fewer than eight correct trials on each side without fixation violations) were excluded from analyses. These sessions were rare and were usually caused by problems with the hardware (e.g., a clogged water-reward valve or a dirty IR-photodetector). We collected ≈145,000 control trials (range [57692 265332]) over ≈450 (range [291640]) sessions from each rat (n = 14) from sessions without intracranial infusions or pre-session anesthesia for a total of 2,057,074 control trials.

## Surgery

Surgical methods were identical to those described previously (*Erlich et al., 2011*). Briefly, rats were anesthetized with isoflurane and placed in a stereotax. The scalp was deflected and the skull was cleaned of tissue and blood. The stereotax was used to mark the locations of craniotomies for the FOF and PPC on the skull. Craniotomies and durotomies were performed and then the skull was coated with a thin coat of C&B Metabond (Parkell Inc., NY). Guide cannula (Plastics One, VA) were lowered to

brain surface with dummy cannula extending 0.5 mm into the brain. The guide cannula were placed and secured to the skull one at a time with a small amount of Absolute Dentin (Parkell Inc., NY). After the three guide cannula were in place (One bilateral FOF cannula and one cannula for each PPC) dental acrylic (Duralay, Reliance Dental Mfg. Co, IL) was then used to cover the skull and further secure the cannula. Rats were given 5 days to recover on free water before resuming training.

## Cannula

Group 1 rats (n = 6) were implanted bilaterally in FOF (+2 AP, ±1.25 ML mm from Bregma) with 22 AWG guide cannula (C232G-2.5, Plastics One, VA) and the medial (3.8 mm posterior, 2.2 mm lateral to Bregma) and lateral (3.8 mm posterior, 3.4 mm lateral to Bregma) PPC with 26 AWG guide cannula (6 cannula per rat total). Group 2 rats (n = 4) were implanted in FOF and in PPC (3.8 mm posterior, ±2.8 mm lateral to Bregma; a total of 4 cannulae per animal) with bilateral 22 AWG guide cannula. Group 3 rats (n = 4) were implanted with bilateral 22 AWG guide cannula in FOF and in PPC (4.5 mm posterior, ±3.0 mm lateral to Bregma; 4 cannulae per animal).

The tip of the guide sat at brain surface and the dummy extended 0.5 mm into cortex. The injector for the 22 AWG guide cannula was a 28 AWG cannula that extended 1.5 mm below the bottom of the guide cannula. The injector for the 26 AWG guide cannula was a 33 AWG cannula that extended 1.5 mm below the bottom of the guide cannula.

## Infusions

In general, infusions were performed once a week with control training days taking place on all other days of the week in order to minimize adaptation to the effects of the muscimol and to have good stable performance in the sessions immediately before infusion sessions. On an infusion day, the rat was placed into an induction chamber with 2% isoflurane, and then transferred to a nose cone with 2% isoflurane for the infusion procedure. Caps and dummy cannula were removed and cleaned. Injectors were placed into the relevant guide cannula and extended 1.5 mm past the end of the guide, into cortex. We used a hamilton syringe connected via tubing filled with mineral oil to the injector to infuse 0.3 µl of muscimol (of various concentrations–see 'Results' and *Figure 3—figure supplement 2*) into cortex. After injection, we left the injector in the brain for 4 min to allow diffusion before removal. After 4 min, cleaned dummies were placed into the guide cannula and capped and the rat was removed from isoflurane. After 30 min of recovery from isoflurane the rat was placed into a behavior box as usual. See *Figure 3—figure supplement 2* for the complete list of all infusion doses, regions, and order for each rat. Previous experiments in rat cortex, performing autoradiographic estimates (*Martin, 1991*), as well as simultaneous muscimol inactivation and recordings (*Krupa et al., 1999*), suggest that at the doses of muscimol we used, the expected area of inactivation would have an approximately ≈1 mm radius for the smallest doses and >3 mm radius for the largest doses.

Since the only difference between left and right infusions (and FOF and PPC infusions) are the location of the infusion any differences in behavior can only be attributed to the infusion and not to handling. For bilateral infusions, we were interested in non-lateralized impairments compared to baseline performance. To rule out the possibility that the handling of rats for the infusion procedure could affect performance we did isoflurane-only sessions where rats were handled as they would be for an infusion (taken to the infusion room, placed into an induction chamber with 2% isoflurane, dummies removed, cleaned and replaced, etc) but given no infusion. These isoflurane-only sessions were used as a baseline to compare with the bilateral infusion sessions.

## Analysis and statistics

All analysis and statistics were computed either in Matlab (version 7 or better, The Mathworks, MA) or R (version 2.15.2, R Foundation for Statistical Computing, Vienna, Austria). Sessions where there were too few trials, or otherwise had problems during the infusions were excluded. We only included the first 250 trials of a session because the effect of muscimol can decrease after a few hours, although the results are robust to including all trials.

The accumulator model uses 9-parameters (described in *Figure 2—figure supplement 1*) to transform the stimulus on each trial (input to the model as the left and right click times) into a probability distribution about the choice of the rat. For example, if for a given set of parameter, the model predicts that trial 1 will result in 75% chance of the rat going right, and the rat in fact went right, that trial would be assigned a likelihood of 0.75. In the case that the rat went left, the trial would be

assigned a likelihood of 0.25. We fit the model assuming that the trials are independent. Therefore, for a model with parameters $\theta$ for all decisions D, the likelihood is given by:

$$P(D|\theta) = \prod_i P\left(d_i | t_{i,R}, t_{i,L}, \theta\right),$$

the product of the likelihoods of the decisions on trial $i$, $d_i$, given the times of the right clicks $t_{i,R}$, times of the left clicks $t_{i,L}$, and the set of 9 parameters, $\theta$. A detailed description of the procedure for fitting the accumulator model can be found in the Modeling Methods section of the supplement of *Brunton et al. (2013)*. For panels A, C, and D in *Figure 2* we first concatenated 47,580 trials across sessions and rats from sessions 1 day before an infusion session. The values of the parameters that maximized $P(D|\theta)$ for these data are described as 'Meta-Rat' in *Table 1*. Since the model is fit to the individual trials, we emphasize that the psychometric, chronometric and reverse correlations plot for a given model are not generated using a curve-fitting procedure for each panel. We also fit each rat individually and show the best-fit parameters for each rat in *Table 1*.

Even at the very easiest trials, rat behavior in our task often asymptotes at $\approx$90% correct. To fit this, the model includes a 'lapse' parameter, which represents a fraction of trials in which subjects will ignore the stimulus and choose randomly. The presence of the lapse parameter also puts a lower bound on the likelihood of any individual trial, and thus no individual trial can dominate the results and the consequent fits of the model.

In *Figures 3, 4E, 6, 8E*, *Figure 2—figure supplement 2A* the psychometric curves were generated by concatenating trial data across sessions for each rat and fitting (using Matlab's nlinfit) a 4-parameter sigmoid as follows:

$$y = y_0 + \frac{a}{1 + e^{\frac{-(x - x + 0)}{b}}}.$$

For these fits, $x$ is the click difference on each trial (#Right Clicks − #Left Clicks), $y$ is 'P(Went Right)', and the four parameters to be fit are: $x_0$, the inflection point of the sigmoid; $b$, the slope of the sigmoid; $y_0$, the minimum 'P(Went Right)'; and $a + y_0$ is the maximum 'P(Went Right)'. These fits are for visualization only.

For chronometric analyses (*Figures 2C*, *4C*, *6D* and *Figure 2—figure supplement 2C*), we concatenated trials for each rat across sessions and binned trials into easy, medium and hard quantiles with equal number of trials based on the relative generative left-right click rate. For each of these three difficulty levels we binned trials by stimulus duration.

The detailed methods of generating the psychophysical reverse correlations (*Figures 2D, 4D, 5D* and *Figure 6—figure supplement 1*) can be found in the *Brunton et al. (2013)*. For this analysis separation of right (red lines) and left (green lines) trials at each time in the trial indicates that there was a difference in local click rate at that time for trials in which the rat responded to the right vs to the left. If rats only used the early clicks for their decision (a primacy strategy), the lines would begin separated and come together. Likewise, if they only used the clicks at the end of the stimulus (a recency strategy) the lines would start together and separate towards the end of the trial. To generate the accumulator model's reverse correlation, each trial was assigned as the left and right trial, according to the model's prediction for that trial. For example, if the model predicted that a trial had a 67% chance of a rightward choice, this trial would contribute 0.67 to the right trials and 0.33 to the left trials psychophysical kernel.

The unilateral infusion experiments are designed specifically to test for lateralized effects. On left and right infusion days animals are handled in the exact same way: taken into the room where the infusions happen, placed under light isoflurane anesthesia, etc (described in detail in the Infusions section of the 'Materials and methods'). The only difference is the side of the infusion. As such, this within-subject design is effective at testing biases to respond toward or away from the side of the infusion. The simplest way to estimate bias resulting from unilateral inactivations is to subtract the contralateral % correct from the ipsilateral % correct for each infusion session. To compute the overall bias we averaged the bias across sessions for each rat and then tested using a $t$-test whether the bias across rats was significantly different from 0. However, this statistical test is conservative, since it collapses across all trials of differing difficulty levels and different sessions. As well, this overall bias measure was inappropriate for testing the effects within each group, since the n per group was low.

To avoid false negative results (type II error), as a more sensitive measure of inactivation effects we used a Generalized Linear Mixed-Model (GLMM) as implemented in the function

'lmer' in package 'lme4' (*Bates and Sarkar, 2007*; *Bates et al., 2007*). For unilateral infusions we specified a mixed-effects model where the rats' choice on each trial was a logistic function of #Right − #Left Clicks (Δ Clicks), infusion side and their interaction as fixed effects. The rat and an interaction of rat, infusion side, and Δ Clicks were modeled as within-subject random effects. The statistic reported in the text for unilateral infusions was the p value for the infusion side fixed effect. The plots in *Figure 3—figure supplement 3* (FOF) and *Figure 7—figure supplement 1* (PPC) show that the model fits for each rat are quite good, and reflect how the random effects of the model allow for each rats' data to be fit, while also finding significant fixed effects. The logistic fit sometimes misses the end point performance, which, for the FOF inactivations, was generally worse than the GLMM fit.

For bilateral infusions we specified a similar model comparing bilateral infusions to isoflurane-only sessions. For these models the relevant statistic was the significance of the interaction between infusion and Δ Clicks, that is, a change in the slope of the logistic. Details of the data and code used to generate and compare the models is described in *Supplementary file 1* 'Using the lme4 package to fit generalized-linear mixed models in R'.

To test whether there were effects of unilateral infusions on response times (RT) on side LED trials, measured as time from go cue to side port response, we took the mean RT for each rat for left and right choices on left and right infusion days. We separated leftward and rightward responses because there can be large differences from rat to rat in the left vs right response times. We then did a repeated-measures ANOVA (Rat X [Left vs Right] X [Ipsi vs Contra]) to test whether there was an effect of muscimol on RTs. Since ANOVA must be balanced, only 4 rats (out of a possible 8 group 2 and 3 rats) performed enough side LED trials from FOF infusions for this analysis. For the PPC infusions 7 of 8 rats performed enough trials to be included.

For the analyses of biases during free-choice sessions we compared the bias (ipsilateral − contralateral % correct) on the infusion day with the control session 1 day before and performed *t*-tests across sessions for each region (PPC or FOF) and trial-type (free-choice, accumulation or side LED). Some free-choice sessions had no side LED trials which is why there are fewer sessions in the *t*-tests for that trial-type. Side LED trials were also not analyzed for sessions if the bias in the side LED trials during the preceding control day was greater than 20%.

To fit the 9-parameter accumulator model to the bilateral FOF infusion data we concatenated all 1809 trials across all rats. We validated the error in these fits by bootstrapping. Specifically, we generated 300 sets of 1809 trials data by randomly sampling with replacement from the original data set. We then fit the 9-parameter model to each of these 300 data sets, generating 300 sets of 9 best-fit parameters. We used this distribution of fits to generate confidence intervals for each of the parameters. In order for a parameter to be considered as significantly different from the control value, the control value from the meta-rat fit had to be outside of the 95% confidence interval of the bootstrapped marginal distribution for that parameter.

To better understand what aspect of the task was impaired by unilateral FOF inactivation we took advantage of the knowledge of the times of all the clicks and the rats' choices using our previous modeling work (*Figure 2—figure supplement 1*, *Brunton et al., 2013*) to fit four constrained Accumulator Models to the unilateral FOF infusion data. For all of these analyses we relabeled the trials for left/right trials to ipsi/contra trials based on the side of the infusion. For the infusion meta-rat we combined all 3836 trials from unilateral FOF sessions. Second, to avoid computing the gradient for a 12-parameter model, we fit constrained Accumulator Models that had one parameter free and the other parameters fixed to the best-fit parameters from the control meta-rat Accumulator Model. The details of the free parameter of each model are described in the results section.

We used MALTAB's built-in Metropolis–Hastings sampler (mhsample.m in the Statistics Toolbox) to sample from the 8-parameter distribution (described in the 'Results'). We generated 4 separate Markov chains of 10,000 samples each, starting from the best-fit control parameters with a burnin of 100. For the proposal distribution we used a uniform distribution with the range selected for each parameter (e.g., a smaller range for input gain than for input noise). We used a thin of 4, based on test runs with known gaussian distributions. *Figure 6—figure supplement 2* shows the samples generated by this sampling process.

To compare models with different numbers of parameters we used two commonly used metrics, the Bayesian and Akaike information criterion (BIC and AIC). The BIC is defined as, $BIC = -2 \times LL + k \times ln(n)$. Where LL is the maximum log likelihood of a model with $k$ free parameters on n data points. The AIC is similarly defined, $AIC = -2 \times LL + 2 \times k$. The main difference being that AIC is not a function of the size

of the data set. Since the cost of additional parameters in AIC is only $2k$, the AIC favors more complicated models compared to BIC.

## Acknowledgements

We thank A Begelfer, K Osorio and J Teran for animal and laboratory support. TDH was supported by National Institutes of Health (NIH) Award Number F32MH098572. CAD was supported by a Howard Hughes Medical Institute International Student Research Fellowship.

## Additional information

### Funding

| Funder | Grant reference | Author |
|---|---|---|
| Howard Hughes Medical Institute (HHMI) | International Student Research fellow | Chunyu A Duan |
| Howard Hughes Medical Institute (HHMI) | Investigator | Carlos D Brody |
| National Institutes of Health (NIH) | F32MH098572 | Timothy D Hanks |

The funders had no role in study design, data collection and interpretation, or the decision to submit the work for publication.

### Author contributions

JCE, Participated in all aspects of this project; BWB, Collaborated with TDH and JCE in adapting the model from Brunton et al. (2013) for this project; CAD, Participated in data collection and provided comments on the manuscript; TDH, Collaborated with BWB and JCE in adapting the model from Brunton et al. (2013) for this project, Acquisition of data, Analysis and interpretation of data, Drafting or revising the article; CDB, Conception and design, Analysis and interpretation of data, Drafting or revising the article

### Author ORCIDs

Jeffrey C Erlich, http://orcid.org/0000-0001-9073-7986
Bingni W Brunton, http://orcid.org/0000-0002-4831-3466
Chunyu A Duan, http://orcid.org/0000-0002-3095-8653

### Ethics

Animal experimentation: This study was performed in strict accordance with the recommendations in the Guide for the Care and Use of Laboratory Animals of the National Institutes of Health. All of the animals were handled according to approved institutional animal care and use committee (IACUC) protocol (#1853) of Princeton University. All surgery was performed under isoflurane anesthesia and analgesics were given for 5 days after surgery, as recommended by the institute veterinarian. Every effort was made to minimize suffering.

## Additional files

### Supplementary file

• Supplementary file 1. Using the lme4 package to fit generalized-liner mixed models in R. This file contains the code (and links to our data) which shows how we used the lme4 package, in R, to fit generalized linear mixed models (GLMM). We also include the output of each of the GLMM we described in the main text. This allows the interested reader to regenerate our main results and also, by providing the data, allows the reader to perform additional statistical tests.

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
