## [Decision Letter]

Thank you for sending your work entitled “Distinct effects of prefrontal and parietal cortex inactivations on an accumulation of evidence task in the rat” for consideration at *eLife*. Your article has been favorably evaluated by Eve Marder (Senior editor) and 4 reviewers, one of whom, Matteo Carandini, is a member of our Board of Reviewing Editors.

The Reviewing editor and the other reviewers discussed their comments before we reached this decision, and the Reviewing editor has assembled the following comments to help you prepare a revised submission.

This is a timely report on a topic of great interest, with impressive behavioral data and analysis. It stands to make a significant contribution to the decision-making literature. The manuscript investigates the causal contributions of rat prefrontal cortex (PFC) and posterior parietal cortex (PPC) in accumulation-based decision making. It provides a strong synthesis of many kinds of inactivations (bilateral vs unilateral, PFC vs PPC, and some combinations) with a detailed computational model of the decision making process. Despite some important caveats (described below) with the inactivation experiments and with their modeling, the experiments make up a very useful data set. Moreover, the follow-up manipulations to test predictions from the results of the analyses (single-sided trials, free choice trials) demonstrate the success of the method and the depth of understanding of the role of these structures.

This approach maximizes what one can learn from the rodent relative to the primate – flexible and creative application of causal techniques, coupled with detailed modeling of the behavior that allows them to transcend some concerns about rodent psychophysics, for example, the high lapse rates. In many regards, this work leads the primate research and will be an important reference point as corresponding investigations of primate PPC and PFC catch up.

The main finding is quite compelling: large effects of PFC inactivation on decision-making, little or no effect of PPC inactivation. These results are not entirely surprising given what we know about the roles of FOF from the Brody lab and others ([24]; Guo et al., 2013; Hanks et al), and given the results seen in PPC with binary decision tasks (Guo et al., 2013). However, the nuances and follow-ups raise the rigor of the work and unpack the mechanism in a satisfying way.

Specifically, the modeling suggests that inactivation of the Frontal Orienting Field (FOF) largely changes the integration time constant, which is a central aspect of the accumulation mechanism – and which is a strong contrast from less-central effects, such as, say, the lapse rate.

The apparent lack of effect in PPC during the evidence accumulation task is intriguing, since PPC has been at the center of research on evidence accumulation in primates. The suggestion that it may not be necessary for performing evidence accumulation is important. However, given the prominence of this finding, one would need to see a bit more characterization of PPC infusions before being confident of these claims. Moreover, as discussed below, these findings need to be better described in the context of the literature, as PPC inactivation does have major effects during different kind of perceptual decisions: those based on vision ([Bibr bib63a]).

*1) Use of muscimol*

The manuscript should explain the advantages of using muscimol inactivation over the better-characterized optogenetic approach used in the same lab (Hanks et al.). Muscimol infusion has much poorer spatial and temporal resolution, and given the depth of these injections (1.5 mm below the cortical surface), one worries that muscimol could be affecting basal ganglia in the FOF infusions (e.g., the single example provided in Figure 3—figure supplement 1, Figure 3—figure supplement 2 and Figure 3—figure supplement 3 seems to show some subcortical spread). These limitations should be discussed.

*2) Incomplete PPC inactivations?*

The paper discusses the lack of consensus regarding the stereotaxic location of PPC, and addresses this by infusing multiple locations in different groups of animals. This is unconvincing, especially given the importance of the negative results in PPC to the main points of the paper. It would be useful to see more anatomical characterization to show that what is being called PPC has the appropriate input characteristics. This could be done fairly easily in a few naïve animals using retrograde tracers showing that their identified region receives inputs from multiple sensory regions, lateral posterior thalamus, prefrontal cortex, and contralateral PPC.

Indeed, a major concern is that PPC inactivations may be only partial, given the large medial-lateral extent of rat PPC (as the paper notes, ∼4mm), the muscimol infusions are likely only partially inactivating PPC, and the lack of effect could be due to compensation by the spared neurons. It is nice that there is a positive control in the 'free choice' experiments, but it is not clear that one can rule out a more subtle influence of PPC in the evidence accumulation task. The experiment one would really like to see is a characterization of the extent of PPC using anatomical tracers, followed by inactivation of the entire region. This would require significantly more work, and may be beyond the scope of a reasonable revision of this paper, but at the very least the paper should address this issue in the Discussion and reduce the strength of the claims accordingly.

*3) Relationship with visual study*

In its current version, the paper states that PPC plays no role in perceptual decisions. A revision should clarify that this applies to ”auditory” perceptual decisions. The recent paper by [Bibr bib63a] shows similar lack of effects on an auditory task, but profound effects in a visual task. The manuscript relates to this study in one paragraph (eighth paragraph of Discussion) to suggest that PPC may be mostly visual rather than auditory, but seems to prefer the rather harsh interpretation that other investigators had results in visual tasks due to experimental error. It is dangerous to criticize others for possible technical errors (muscimol spreading unintentionally far) when one's paper may suffer from the same limitations. It would be interesting to see a more balanced discussion of those results, perhaps starting in the Introduction. For instance, there may be some tension about whether those results point to an evidence accumulation deficit for visual information, or just a visual deficit. More generally, this paper should make sure not to overreach in its conclusions about the role of PPC in all sorts of perceptual decisions, and limit itself to auditory ones.

*4) Fitting unilateral FOF infusion data*

When fitting the unilateral FOF infusion data, the current approach seems to be to allow only the bias parameters to vary, while constraining the remaining parameters to the values obtained when fitting control data. However, bilateral FOF inactivation had a large effect on several of those fixed parameters (accumulation time constant, sensory noise, etc.). These parameters (particularly accumulation time constant) could reasonably be expected to change during unilateral FOF inactivation. Thus, assuming incorrect “control” values for these parameters could lead to incorrect fitting of the bias parameters.

It seems that any parameters likely to be affected by unilateral FOF inactivation (e.g., those strongly affected by bilateral inactivation) should be free to vary. This could be done through nested model fits, and given that the model fits are based on likelihoods, statistical nested-model comparison could be performed (e.g., likelihood ratio test) in addition to direct visualization of the likelihood surfaces.

An additional suggestion is to repeat the analysis (testing four versions of bias parameters) but do it multiple times, each time leaving eight of the nine original parameters fixed and one unconstrained. If indeed each of these nine tests shows that the unconstrained original parameter took the same value as in the control data, then the method of fixing the nine would be more justified.

Finally, when comparing four alternative explanations for unilateral FOF results, is there any way to be sure that a local maximum has not been found in the parameter space? For instance, perhaps given one solution for the original 9 model parameters, the Post-Cat-Bias manipulation accounts for unilateral FOF inactivations best, but given another solution the Input Gain would fit best. In the original paper about the model the authors stated that the model is not proven concave, so in a case like this it seems like one must worry that a local maximum may have been found (or, even if a global maximum was found, one may consider that another peak of comparable likelihood exists).

*5) Distinguishing effects on lapse rate and accumulation time*

Another concern regards the possibility of bolstering the model-based inferences from the behavioral data. The ability to distinguish between effects on lapse rate and accumulation time constant is critical to this paper. There are two related components one would like to see strengthened. First, Figure 4 would be bolstered by an elaboration of the analysis shown in panel D. The reverse-correlation analysis compares the data and the model (the latter, relying on an effect on the time-constant of accumulation). Consideration of how the rev-co kernel would look for the alternate model (lapse rate) may provide a complementary perspective on why this model is insufficient, perhaps broadening the support beyond the likelihood-based metric.

A related concern regards likelihood-based fits. These are of course statistically principled, but in practice the metric can explode at the extremes: the most common instance is for binomial data with 0% or 100% accuracy (i.e., the extremes of a nice psychometric function). Now, these extremes may not be at play in the key analysis here, so the concern is not particularly strong, and in general the notion of showing the likelihood space as a function of parameters is powerful and richer than the usual reliance on p-values. But fleshing out whether (and if so, how), extremes were handled would be helpful.

*6) Unclear results of the side LED experiments*

There seems to be a disagreement between the average data obtained in the side LED experiments (Figure 3), and the data in individual trials (Figure 3—figure supplement 3). This is hard to tell, because Figure 3—figure supplement 3 is insufficiently labelled. Specifically, are the two end points in these curves the side LED trials? This in unlabeled but seems to be true since there are eight points per plot, matching the six click differences + two side LED types from main Figure 3. If so, then one can hardly understand how the side LED trials are unaffected in the average (Figure 3) because it's clear from looking at individuals (Figure 3—figure supplement 3) that in almost all cases these trials are massively shifted. If the data plotted in Figure 3—figure supplement 3 are only click trials, then plotting the side LED trials as well would be helpful, and also it would be helpful to explain why eight click-rate-differences are plotted for each individual in Figure 3—figure supplement 3 but only six in main Figure 3. It is important to clarify this because much of the interpretation rests on the result that unilateral FOF inactivations do not influence performance on side LED trials (to rule out a motor explanation).

*7) Unsatisfactory model comparison*

In the comparison of Input Gain and Post-Categorization-Bias models, a look at the data suggest the paper dismisses the input gain model too quickly. First, it seems to fit the reverse correlation plots slightly better (while fitting the psychometric functions slightly worse); Figure 6—figure supplement 1. Second, there is no metric for interpreting how impactful (on behavioral performance and/or fit quality) a shift of 0.4 in post-cat-bias relative to a shift of 0.3 in input gain might be (visually estimating 0.3 from Figure 6). Similarly, the statement that post-cat-bias is “50 log-units better” than input gain sounds impressive but is difficult to interpret. Is it possible to do a cross-validation test? Using the parameters fit [with the post-cat-bias, input gain, or both] from 90% of the trials, predict behavior on the other 10% (or leave-one-out, or however one wishes to do this)? This would be clearer to interpret and would also allow for the comparison of the model including both parameter shifts to the models including only one. Finally, if each model is fit to each animal (or session), then it seems that Figure 6 can have error bars, which would be further helpful. Somewhat related to this: Can the post-cat-bias be understood in the same way as the bilateral effects, namely as a shortened memory? That is, even in the single-sided trials, perhaps the rat forgets his choice in the quarter second between the go cue and the start of his movement? I.e. even this massive click train still decays with tau=-0.24 s.

*8) Unclear design of the free-choice trials*

On the free-choice trials, it is unclear whether muscimol was injected into a side of the brain specifically chosen to be contralateral to the bias from the previous day, or whether it was injected into a random side. If random, why is there such a strong contralateral bias in control? If chosen based on previous day's bias, this effect could reflect regression to the mean; indeed the muscimol points in Figure 9 appear to have somewhat close to zero mean. An appropriate control would be to make the same selection of which side is defined as ipsilateral, but then do a control (“isoflurane”) injection session instead of muscimol. (The legend seems to indicate that the points labeled ”Control“ in Figure 9 are not control injections but rather data from the previous day relative to the muscimol points; if this was an incorrect understanding then the legend could be clarified.)

---

## [Author Response]

We believe we have sufficiently addressed all the issues brought up by the reviewers. First, we have edited the text of the manuscript to better reflect the conclusions and limitations of our results. Second, we have updated figures and captions to avoid confusion and address the potential confounds that reviewers addressed. Finally, we did extensive additional analyses to support our model-based conclusions about the role of the FOF. All of our additional analyses support our original claims, and we are grateful to the reviewers for helping us to make the paper stronger.

1) Use of muscimol

*The manuscript should explain the advantages of using muscimol inactivation over the better-characterized optogenetic approach used in the same lab (Hanks et al.). Muscimol infusion has much poorer spatial and temporal resolution, and given the depth of these injections (1.5 mm below the cortical surface), one worries that muscimol could be affecting basal ganglia in the FOF infusions (e.g., the single example provided in*
Figure 3—figure supplement 1, Figure 3—figure supplement 2 and Figure 3—figure supplement 3
*seems to show some subcortical spread). These limitations should be discussed*.

Muscimol versus halorhodopsin: Thank you for the excellent question. We should have described the rationale for our choice in the manuscript. When we began using halorhodopsin in the lab, we thought it would entirely replace our use of muscimol. However, we came to understand that instead of one technique replacing the other, the two complement each other. The reviewers’ question made us further realize that other labs facing a similar choice between inactivation techniques might find our rationale to be of interest. We thus now write in the manuscript:

“Although optogenetics allows high temporal resolution inactivation, its radius of effect in our hands is only ∼750 um (Hanks, Kopec, et al. 2015). In contrast, inactivating larger regions is much more readily achieved with muscimol, for which the radius of inactivation can be increased simply by increasing the infusion dose. In addition, muscimol directly inactivates all neurons within its radius of effect, not only infected neurons. Moreover, in some tasks, including the Poisson Clicks task, we find that the behavioral effect of optogenetic silencing begins to decay after a few weeks, while the effect of muscimol is stable. This stability in particular was essential for the current study, which used within-subject manipulations over hundreds of days. Optogenetic and pharmacological silencing therefore have complementary advantages and disadvantages. Here we focused on pharmacological inactivation.”

Subcortical spread of muscimol: There is little risk of muscimol spread from the FOF into the basal ganglia. We should point out that the thick white matter bundle (∼ 1 mm) separating FOF from the striatum below provides an effective barrier to the spread of muscimol in the ventral direction. We should also emphasize that Figure 3—figure supplement 1, Figure 3—figure supplement 2 and Figure 3—figure supplement 3, referred to by the reviewers, is not an image of muscimol spread, but of an anatomical tracer infused into the FOF cannula of one animal. This tracer (cholera-toxin-B conjugated to a fluorophore) is mostly retrograde but also has some anterograde transport. Thus, in contrast to muscimol, which will be blocked by greasy white matter, the CTB will follow axons projecting into the white matter. What is seen in Figure 3—figure supplement 1, Figure 3—figure supplement 2 and Figure 3—figure supplement 3 is CTB going into the most superficial part of the white matter bundle. We apologize for making it easy to confuse CTB spread with muscimol spread, and we have now made sure to clarify and emphasize the difference in both Figure 3—figure supplement 1, Figure 3—figure supplement 2 and Figure 3—figure supplement 3 and its caption.

Other spread of muscimol: We wholly agree that spread of muscimol is important to discuss. We discuss spread from FOF to adjacent cortical regions in the Discussion (from second paragraph of subsection headed “Role of FOF”); we have now added a sentence to that section explaining that the white matter blocks spread ventrally into the basal ganglia. We also discussed issues of spread of muscimol from PPC in subsection headed “Role of PPC”(and see response below).

2) Incomplete PPC inactivations?

*The paper discusses the lack of consensus regarding the stereotaxic location of PPC, and addresses this by infusing multiple locations in different groups of animals. This is unconvincing, especially given the importance of the negative results in PPC to the main points of the paper. It would be useful to see more anatomical characterization to show that what is being called PPC has the appropriate input characteristics. This could be done fairly easily in a few naïve animals using retrograde tracers showing that their identified region receives inputs from multiple sensory regions, lateral posterior thalamus, prefrontal cortex, and contralateral PPC*.

Our bregma coordinates receive input from regions that match those expected from PPC. Although we used bregma coordinates routinely described as “PPC” in the rat literature (e.g., [59]; [Bibr bib63a]), we agree that that anatomical data demonstrating that these coordinates have the connectivity one would expect from PPC would be very appropriate. A recent study from the McNaghuton lab provides these data (Wilber, A. A., Clark, B. J. et al., Cortical connectivity maps reveal anatomically distinct areas in the parietal cortex of the rat. Frontiers in Neural Circuits*,* January 2015). The authors used several tracing techniques to examine the inputs to the rat PPC, and report that the bregma coordinates we used receive input from the expected regions, including the lateral posterior thalamus, multiple sensory cortices, prefrontal cortex, contralateral PPC, and FOF (which is a subregion of M2). Interestingly, in their Figure 6, they should that only the region that we have targeted as PPC (which they consider “rostral” parietal cortex) has input from FOF. We have updated the manuscript in several places to include this important reference.

*Indeed, a major concern is that PPC inactivations may be only partial, given the large medial-lateral extent of rat PPC (as the paper notes, ∼4mm), the muscimol infusions are likely only partially inactivating PPC, and the lack of effect could be due to compensation by the spared neurons. It is nice that there is a positive control in the 'free choice' experiments, but it is not clear that one can rule out a more subtle influence of PPC in the evidence accumulation task*.

Even our largest inactivations may be only partial. Based on our largest doses and the existing literature that has examined the spread of muscimol in rat cortex (Krupa, 1999) we believe that our largest doses should cover the entire PPC. But as the reviewers point out, it nevertheless remains possible that there might be some spared neurons in the PPC that compensate for the inactivated neurons. However, even if this were true, we would expect a dose-response relationship for unilateral muscimol PPC and bias magnitude. No such relationship is seen for accumulation trials (square yellow data points, Figure 8). Furthermore, the accumulation task contains trials that are at psychophysical threshold (i.e. 50% performance). If unilateral PPC inactivations had a behavioral effect, we would have expected it to be revealed on those trials. On balance, then, we feel the evidence leans towards a lack of effect from unilateral PPC inactivations alone. But we acknowledge in the Discussion that we cannot completely rule out the possibility that unilateral inactivations could produce an effect. Note that lack of an effect after unilateral PPC inactivations alone is different, as we discuss in the next paragraph, from a subtle role for PPC in the accumulation task.

We agree, and the data indeed show, that PPC has a subtle but real effect on the accumulation task. The reviewers suggest that we acknowledge that there may be a subtle influence of the PPC in the accumulation task. We very much agree, and that is what we would have liked to portray. In the abstract, we use the term “minimal” for the role of PPC, which seems close in spirit to “subtle”. Nevertheless, we fear that our original writing may have been too strong, and may have instead conveyed a “no PPC effect whatsoever” message. If so, that was our mistake, and we thank the reviewers for pushing us into making sure to correct it.

The evidence, included in the original submission, for a subtle but significant effect of PPC inactivations on the accumulation task comes from two findings: (1) there was a weak, but statistically significant effect of 150ng of muscimol bilaterally infused into PPC (Figure 7). (2) unilateral inactivation of 300ng of muscimol into the PPC during simultaneous bilateral inactivation of the FOF produced a small, but clear and measurable bias (Figure 9). We note that both of these effects were produced at doses of muscimol substantially lower than the higher 600ng and 2500ng doses used in the unilateral PPC experiments. This suggests to us that the lack of effects that we saw in the unilateral PPC experiments were not due to spared neurons. Nonetheless, we have softened our language in the Discussion ( paragraph three of subsection headed “Role of PPC”).

*The experiment one would really like to see is a characterization of the extent of PPC using anatomical tracers, followed by inactivation of the entire region. This would require significantly more work, and may be beyond the scope of a reasonable revision of this paper, but at the very least the paper should address this issue in the Discussion and reduce the strength of the claims accordingly*.

We agree that that would indeed require far more work than would fit within a revision, and we thank the reviewers for the opportunity to address the issue in the Discussion and reduce the strength of the claims appropriately. See subsection headed “Role of PPC”.

3) Relationship with visual study

*In its current version, the paper states that PPC plays no role in perceptual decisions. A revision should clarify that this applies to ”auditory” perceptual decisions. The recent paper by*
[Bibr bib63a]
*shows similar lack of effects on an auditory task, but profound effects in a visual task. The manuscript relates to this study in one paragraph (eighth paragraph of Discussion) to suggest that PPC may be mostly visual rather than auditory, but seems to prefer the rather harsh interpretation that other investigators had results in visual tasks due to experimental error. It is dangerous to criticize others for possible technical errors (muscimol spreading unintentionally far) when one's paper may suffer from the same limitations. It would be interesting to see a more balanced discussion of those results, perhaps starting in the Introduction. For instance, there may be some tension about whether those results point to an evidence accumulation deficit for visual information, or just a visual deficit. More generally, this paper should make sure not to overreach in its conclusions about the role of PPC in all sorts of perceptual decisions, and limit itself to auditory ones*.

We agree that the bulk of our data, from an auditory accumulation of evidence task, provides evidence only about accumulation of auditory evidence, and that we should correct our writing wherever it gives a misleading impression. Consequently, throughout the manuscript, beginning with and including the abstract, we have endeavored to replace statements about “evidence accumulation” with statements about “auditory evidence accumulation”. We hope this will clarify the issue, and thank the reviewers for pointing out the problem.

We also fully agree that issues of muscimol spillover could apply to our own study. In our original submission we tried to discuss these issues directly, and specifically with respect to our own data and conclusions. We have kept all those sections in this resubmission. If the reviewers think that those sections are insufficient, and that a more in-depth discussion of how the issue affects our own data is necessary, we would be happy to expand those sections.

With respect to the Raposo et al. visual+auditory study, our concerns regarding spillover are a reflection of concerns that we have with data from our own lab: we have recently begun using a visual variant of the Poisson Clicks task (Scott et al., SFN 2014, Constantinople et al., SFN 2014), and following Raposo et al., we find, as they did, that infusing muscimol into “standard” PPC coordinates impairs rats performing the visual task even while it leaves rats performing the auditory task intact (Scott et al., unpublished preliminary data). Nevertheless, the literature regarding the location of PPC seems to us unclear with respect to the PPC/visual cortex border. This has made us very reluctant to firmly conclude that our inactivation at PPC coordinates avoids inactivation of visual cortex, and our concerns with our own data seem to us to also apply to the data in Raposo et al.

Raposo et al. cited a single study (Reep at al. 1994) to define the posterior border of PPC at -5.0 mm from Bregma. This definition was critical to their conclusion that their inactivations, which were centered at – 3.8mm from Bregma, had not gone past -5.0, and had thus not invaded visual cortex. However, based on and specifically citing the very same data (originally reported in Chandler, King, Corwin, and Reep 1992) that led [69] to define PPC as extending to -5.0mm from Bregma, Corwin and Reep, the very same authors, four years later defined PPC as extending only to -4.4mm from Bregma ([18], page 90). In other words, there is substantial uncertainty in the location of the border even within a single data set from a single set of authors. Using a -4.4mm definition, the Raposo et al. inactivations would have extended well into visual cortex. Our point is not that the Raposo et al. inactivations necessarily extended into visual cortex. Our point is that our reading of the current state of the literature, suggests that it is very difficult to rule out the possibility that they did.

Another, less direct, example of the lack of consensus or precision in the literature regarding the PPC/visual cortex border is that the standard rat atlas (Paxinos and Watson, 6th edition), places the PPC/V2 border even *more* anterior, at ∼-4.14mm from Bregma, making the problem even more acute.

This problem is much less of a concern for the auditory tasks, since we do not expect visual cortex to play a major role in those.

We also believe that it is interesting to maintain a comparison and a dialogue with results from similar primate research. That motivates us to consider possible interpretations that reconcile results across species, which is what the spillover hypothesis would achieve. All this said, we certainly do not intend to be “harsh” with respect to the Raposo et al. study (to quote the word used by the reviewers). We are strong supporters of that lab’s line of work, and we thank the reviewers for pointing out that our writing came across as harsh, thus giving us the opportunity to correct that. We have now tried, in that discussion section, to both clarify our thinking about the uncertainty in the location of the border, and to soften our language, clearly labeling the spillover possibility as intellectually appealing because it reconciles results across species, but nevertheless undoubtedly speculative. That section now reads

“[Bibr bib63a] reported an impairment after rat PPC inactivations in a visual, but not in an auditory, version of a closely related task. This contrasts with a preliminary report from Yates et al. that suggested no effect from primate PPC inactivation in a visual accumulation of evidence task (90), as well as with multiple reports of no effect from primate PPC inactivation in visual memory-guided saccade tasks (14; 85; 50; 89). […] While such a reconciliation of results across multiple species could have some intellectual appeal, we emphasize that experiments that would either clearly support or rule out this possibility remain to be done.”

We hope this sounds much less harsh, while still laying out all the issues for discussion. We also hope that highlighting the uncertainty in the PPC/visual cortex border may help motivate further studies to pin down its location, which could have significant impact on future studies that attempt to target PPC specifically.

4) Fitting unilateral FOF infusion data

*When fitting the unilateral FOF infusion data, the current approach seems to be to allow only the bias parameters to vary, while constraining the remaining parameters to the values obtained when fitting control data. However, bilateral FOF inactivation had a large effect on several of those fixed parameters (accumulation time constant, sensory noise, etc.). These parameters (particularly accumulation time constant) could reasonably be expected to change during unilateral FOF inactivation. Thus, assuming incorrect “control” values for these parameters could lead to incorrect fitting of the bias parameters*.

Motivated to address the reviewers concerns we used the Metropolis-Hastings algorithm to sample from an 8-parameter model that included the 4 bias parameters, as well as 4-parameters of interest from the original model (accumulation time constant, accumulator noise, sensory noise and lapse). This technique is computationally expensive and not guaranteed to find the global optimum. Nonetheless, allowing all 8 parameters to vary simultaneously, the post-categorization parameter still had the biggest change, small changes in other parameters that gave a higher likelihood than the 1-parameter post-categorization model. However, the improvement was small and using bayesian information criteria we conclude that the 1-parameter model is better. Please see the section on modeling the unilateral FOF data for more details.

*Finally, when comparing four alternative explanations for unilateral FOF results, is there any way to be sure that a local maximum has not been found in the parameter space?*

We had the same concern as the reviewer and had previously examined the shape of the likelihood distribution for each of the 1-parameter models. As it turns out, there is a single maximum for each of the 1-parameter models. We now include those plots in Figure 6—figure supplement 1.

5) Distinguishing effects on lapse rate and accumulation time

*Another concern regards the possibility of bolstering the model-based inferences from the behavioral data. The ability to distinguish between effects on lapse rate and accumulation time constant is critical to this paper. There are two related components one would like to see strengthened. First,*
Figure 4
*would be bolstered by an elaboration of the analysis shown in panel D. The reverse-correlation analysis compares the data and the model (the latter, relying on an effect on the time-constant of accumulation). Consideration of how the rev-co kernel would look for the alternate model (lapse rate) may provide a complementary perspective on why this model is insufficient, perhaps broadening the support beyond the likelihood-based metric*.

The lapse model is used as an example of how the psychometric curve is a narrow lens which which to examine deficits, compared to the full model. We could also have used increases in accumulator noise, sensory noise, or initial noise to generate a similar psychometric curve. As such, and motivated by the concerns of the reviewer regarding the reliability of the bilateral FOF fits, we have added several extra model comparison analyses to the paper. We have estimated the confidence intervals of the parameters by resampling the bilateral FOF trials and fitting the model to this resampled data. Given our current computational resources we have managed to fit the model to 300 resamples. This analysis supported our existing conclusion - the time-constant of accumulation became significantly leaky after bilateral FOF inactivation. Please see the Results section on bilateral FOF modeling for more details (subsection headed “Bilateral FOF inactivations reduce the subject’s accumulation time constant”) and the results of the resampling ([Supplementary-material SD1-data]).

*A related concern regards likelihood-based fits. These are of course statistically principled, but in practice the metric can explode at the extremes: the most common instance is for binomial data with 0% or 100% accuracy (i.e., the extremes of a nice psychometric function). Now, these extremes may not be at play in the key analysis here, so the concern is not particularly strong, and in general the notion of showing the likelihood space as a function of parameters is powerful and richer than the usual reliance on p-values. But fleshing out whether (and if so, how), extremes were handled would be helpful*.

Indeed, we encountered this very issue of fitting extremes when developing what later became the [7] model, which we use heavily in the current manuscript. As the reviewer points out, if the model predicts that a particular choice in one trial will occur with near 100% certainty, and the rat makes the opposite choice, that the single trial will have an infinite negative log likelihood. That single trial could thus, on its own, completely skew the results. To address this issue, the model includes a “lapse” parameter, a fraction of trials in which the animal appears to ignore the stimuli and performs randomly. At the very easiest trials, rat behavior in our task seems to asymptote at 90% correct, so the lapse parameter typically has values of ∼ 0.1. This puts a bound on how low the likelihood of a trial can be. For example, with a lapse parameter of 0.1, the log likelihood will never be less than log(0.1), and thus no single trial can dominate the results.

We now write, in the “Analysis and Statistics” section of the Methods,

Even at the very easiest trials, rat behavior in our task often asymptotes at ∼90% correct. To fit this, the model includes a “lapse” parameter, which represents a fraction of trials in which subjects will ignore the stimulus and choose randomly. The presence of the lapse parameter also puts a lower bound on the likelihood of any individual trial, and thus no individual trial can dominate the results and the consequent fits of the model.

6) Unclear results of the side LED experiments

*There seems to be a disagreement between the average data obtained in the side LED experiments (*Figure 3*), and the data in individual trials (*Figure 3—figure supplement 3*). This is hard to tell, because*
Figure 3—figure supplement 3
*is insufficiently labelled. Specifically, are the two end points in these curves the side LED trials? This in unlabeled but seems to be true since there are eight points per plot, matching the six click differences + two side LED types from main*
Figure 3*. If so, then one can hardly understand how the side LED trials are unaffected in the average (*Figure 3*) because it's clear from looking at individuals (*Figure 3—figure supplement 3*) that in almost all cases these trials are massively shifted. If the data plotted in*
Figure 3—figure supplement 3
*are only click trials, then plotting the side LED trials as well would be helpful, and also it would be helpful to explain why eight click-rate-differences are plotted for each individual in*
Figure 3—figure supplement 3
*but only six in main*
Figure 3*. It is important to clarify this because much of the interpretation rests on the result that unilateral FOF inactivations do not influence performance on side LED trials (to rule out a motor explanation)*.

We apologize to the reviewer for the confusion. There are no side LED trials in Figure 3—figure supplement 3 (or any of the supplements showing the GLMM fits). We have edited the figure caption to make this more clear. These supplemental figures serve two purposes. 1) They show the reliability of the muscimol inactivation across rats on the Clicks task. 2) They show the logistic GLMM fits. Because those plots show data and also the logistic “fit” which visualized the GLMM we think having the side LED trials there would be confusing. The choice of eight vs. six points is not important since those points are for visualization and all statistics were done on unbinned data.

7) Unsatisfactory model comparison

*In the comparison of Input Gain and Post-Categorization-Bias models, a look at the data suggest the paper dismisses the input gain model too quickly. First, it seems to fit the reverse correlation plots slightly better (while fitting the psychometric functions slightly worse);*
Figure 6—figure supplement 1*. Second, there is no metric for interpreting how impactful (on behavioral performance and/or fit quality) a shift of 0.4 in post-cat-bias relative to a shift of 0.3 in input gain might be (visually estimating 0.3 from*
Figure 6*). Similarly, the statement that post-cat-bias is “50 log-units better” than input gain sounds impressive but is difficult to interpret. Is it possible to do a cross-validation test? Using the parameters fit [with the post-cat-bias, input gain, or both] from 90% of the trials, predict behavior on the other 10% (or leave-one-out, or however one wishes to do this)? This would be clearer to interpret and would also allow for the comparison of the model including both parameter shifts to the models including only one. Finally, if each model is fit to each animal (or session), then it seems that*
Figure 6
*can have error bars, which would be further helpful. Somewhat related to this: Can the post-cat-bias be understood in the same way as the bilateral effects, namely as a shortened memory? That is, even in the single-sided trials, perhaps the rat forgets his choice in the quarter second between the go cue and the start of his movement? I.e. even this massive click train still decays with tau=-0.24 s*.

We performed cross-validation, leaving out individual sessions. We described this in the caption of a figure, which we realize was easy to miss. We have moved the description of the cross-validation to the main text. In addition, in response to this comment and also major comment #4 we have performed new analyses using the Metropolis-Hastings sampler to further examine the unilateral FOF model fits. Although there was a shift toward negative taus, this was not significant. This new analysis is described starting in the fifth paragraph of the subsection headed “Unilateral FOF inactivations produce a post-categorization bias”.

8) Unclear design of the free-choice trials

*On the free-choice trials, it is unclear whether muscimol was injected into a side of the brain specifically chosen to be contralateral to the bias from the previous day, or whether it was injected into a random side. If random, why is there such a strong contralateral bias in control? If chosen based on previous day's bias, this effect could reflect regression to the mean; indeed the muscimol points in*
Figure 9
*appear to have somewhat close to zero mean. An appropriate control would be to make the same selection of which side is defined as ipsilateral, but then do a control (“isoflurane”) injection session instead of muscimol. (The legend seems to indicate that the points labeled ”Control“ in*
Figure 9
*are not control injections but rather data from the previous day relative to the muscimol points; if this was an incorrect understanding then the legend could be clarified*.*)*

We apologize to the reviewers for the confusion. Infusions were done on both sides, but a session where all the infusions were done to “push” the subjects away from their biases was shown as an extreme example. Since the innate biases are quite large, it is hard to push subjects much in the direction of their existing biases. However, the population panels are based on infusions on both sides and the results are consistent. In order to show this more clearly we have added the day after the infusion to the example session. The infusion day is significantly different from both the day before and the day after. If the finding were simply regressing toward the mean, the bias on the following day would not be expected to return to the control levels of bias.